

# Calibration and validation of the Dynamic Wake Meandering model Part I: Bayesian estimation of model parameters using SpinnerLidar-derived wake characteristics

Davide Conti[1], Nikolay Dimitrov[1], Alfredo Peña[1], and Thomas Herges[2]

[1]Department of Wind Energy, Technical University of Denmark, Frederiksborgvej 399, 4000 Roskilde, Denmark
[2]Sandia National Laboratories, Albuquerque, New Mexico, 87123, USA

**Correspondence:** Davide Conti (davcon@dtu.dk)

**Abstract.** In this first part of two, we study the calibration of the Dynamic Wake Meandering (DWM) model using high spatial and temporal resolution SpinnerLidar measurements of the wake field collected at the Scaled Wind Technology Facility (SWiFT) located in Lubbock, Texas. We derive two-dimensional wake flow characteristics including wake deficit, wake turbulence and wake meandering from the lidar observations under different atmospheric stability conditions, inflow wind speeds and downstream distances up to five-rotor diameters. We then apply Bayesian inference to obtain a probabilistic calibration of the DWM model, where the resulting joint distribution of parameters allows both for model implementation and uncertainty assessment. We validate the resulting fully-resolved wake field predictions against the lidar measurements and discuss the most critical sources of uncertainty. The results indicate that the DWM model can accurately predict the mean wind velocity and turbulence fields in the far wake region beyond four rotor diameters, as long as properly-calibrated parameters are used and wake meandering time series are accurately replicated. We demonstrate that the current DWM-model parameters in the IEC standard lead to conservative wake deficit predictions. Finally, we provide practical recommendations for reliable calibration procedures.

## 1 Introduction

Wake effects are perceived as one of the largest sources of uncertainty in energy production and load estimates of onshore and offshore wind farms (Walker et al., 2016). Within an iterative design process, and/or optimization study, wake effects on aeroelastic turbine responses are predicted using engineering wake models, e.g., the Dynamic Wake Meandering (DWM) (Madsen et al., 2010) and Frandsen (Frandsen, 2007) models, which can be used within simple and fast design tools (Braunbehrens and Segalini, 2019). Their main limitation is their reduced ability to fully resolve the turbulence structures of the wake field, which often leads to an inaccurate representation of the flow field, and biased power and load predictions. To minimize the modelling uncertainty, it is a common practice to calibrate engineering wake models using field measurements, when available, or using higher-fidelity simulations like computational fluid dynamics (CFD).

Wind lidars have become popular for studying wind turbine wakes due to their higher spatial resolution and ease of installation compared to traditional anemometers mounted on meteorological masts (Machefaux et al., 2016). The use of lidar





measurements to calibrate low-order wake models has already been successfully adopted (Trabucchi et al., 2017; Reinwardt
et al., 2020; Zhan et al., 2020a). Although high-quality lidar observations of the wake field are available (Käsler et al., 2010;
Iungo et al., 2013; Aitken et al., 2014), the spatial and temporal resolution required to characterize wake deficit, wake tur-
bulence and meandering characteristics are rarely achieved. Such resolution is a key characteristic for the development and
evaluation of dynamic wake models.

The SWiFT experiment, conducted at the Sandia National Laboratories between 2016 and 2017 (Herges et al., 2017, 2018;
Herges and Keyantuo, 2019), provides a fairly complete and suitable dataset for the calibration and evaluation of wake models
(Doubrawa et al., 2019; Conti et al., 2020a; Doubrawa et al., 2020). The SWiFT dataset consists of concurrent measurements
of inflow conditions from a heavily instrumented meteorological mast, high spatial and temporal resolution measurements of a
single wake flow field behind a turbine from a nacelle-mounted SpinnerLidar (Peña et al., 2019), and power and load measure-
ments from a second turbine operating in the waked field. The detailed instrumentation of the site allows the investigation of
the wake field variability under different atmospheric stability conditions, as well as the analysis of the wake-induced effects
on the waked-turbine operation (i.e., power and load predictions).

Here, we analyse the SWiFT dataset aiming at calibrating and evaluating the DWM model. This model is recommended
in the IEC 61400-1 standard (IEC, 2019) for the purpose of wind turbine and wind farm design certification, and it is widely
used in load assessments under wake conditions (Larsen et al., 2013; Galinos et al., 2016; Reinwardt et al., 2018; Dimitrov,
2019; Reinwardt et al., 2020). The DWM model simulates wind field time series and is divided into three parts: a wake deficit
component, which simulates the velocity deficit, a wake-added turbulence component, and a wake meandering component,
which is a stochastic meandering process. These three components are presumed to affect wind turbine load responses mostly
(Keck et al., 2012; Galinos et al., 2016; Larsen et al., 2013; Dimitrov, 2019). Although several studies have demonstrated the
superior performance of the DWM model compared to other engineering wake models that only predict steady wake features
(Larsen et al., 2013; Reinwardt et al., 2018; Thomsen et al., 2007), the accuracy of both the DWM-simulated wake flow fields,
and the predicted power and loads is still under judgement.

## 1.1   A review on the DWM model

The underlying hypothesis of the DWM model is to consider the wake as a passive tracer of the large incoming turbulence
structures. The so-called *split in scales* assumption (Larsen et al., 2008) states that the large-scale turbulent eddies contained in
the atmospheric boundary layer are the main drivers of the wake meandering, whereas the smaller turbulent eddies govern the
wake deficit evolution downstream of the rotor. Further, wake deficits from upstream turbines are transported in the streamwise
direction, assuming Taylor's hypothesis of frozen turbulence (Larsen et al., 2015). This set of assumptions allows decoupling
the wake deficit and *wake-added turbulence* formulations from the wake meandering process (Larsen et al., 2007). Therefore,
the three components of the DWM model can be computed separately and successively superimposed on turbulence fields to
generate wake time series, which can be used as inputs to aeroelastic simulations (Larsen et al., 2013; Keck et al., 2014a).

The wake deficit formulation of the DWM model is mainly based on the work of Ainslie (1987) and solves the axisymmetric
Navier-Stokes (N-S) equations with an eddy viscosity term and a set of calibration parameters. Initially, the DWM model was



calibrated with CFD simulations performed by Madsen et al. (2010). Keck et al. (2012) derived a two-dimensional model of the eddy viscosity term, and updated the calibration parameters based on CFD simulations. Larsen et al. (2013) found that the calibration parameters of the former two studies were not suitable for predicting power and loads at the Egmond aan Zee offshore wind farm. To match the measured power, they introduced an artificial filtering function in the eddy viscosity term and re-calibrated the deficit model; however, this calibration was not based on the spatial description of the wake flow field but on power production data. The eddy viscosity model to predict velocity deficits in the current IEC standard (IEC, 2019) is inspired by the work of Larsen et al. (2013).

Keck et al. (2014a, 2015) proposed a correction factor to the eddy viscosity term, which includes the effects of atmospheric stability and shear on the turbulence mixing occurring in the wake and re-calibrated the model parameters. Although these improvements were verified against large-eddy simulations (LES), the influence of atmospheric stability on the wake deficit evolution was hardly observed during a lidar campaign (Machefaux et al., 2016; Larsen et al., 2015), in which it was argued that atmospheric stability affects to a large extent the meandering process. A load validation study using the DWM model with calibrated parameters from both Madsen et al. (2010), Keck et al. (2012) and the IEC standard (IEC, 2019) was conducted by Reinwardt et al. (2018), who collected load measurements at the ECN Wind turbine test site in Germany, and at DTU test site in Høvsøre in Denmark. They found fatigue load biases within the range 11–15% for the tower bottom and 8–21% for the blade-root flapwise bending moments. Reinwardt et al. (2020) derived a new set of calibration parameters based on full-field lidar observations of the wake field from a wind farm in the Southeast of Hamburg, Germany. They demonstrated that improved wake deficit predictions can be obtained by calibrating the DWM model with nacelle-mounted lidars.

The fidelity of the simulated wake meandering dynamics also affects the accuracy of load predictions (Larsen et al., 2013; Conti et al., 2020c). Modeling of the meandering process relies on a suitable stochastic turbulence field and definition of the *large-scale* turbulence structures. Larsen et al. (2008) and Trujillo et al. (2011) demonstrated that the *large-scale* eddies can be extracted from the incoming atmospheric turbulence field from local mast measurements . Alternatively, the wake meandering process can be simulated through synthetic wind fields generated using stochastic turbulence models (i.a., the turbulence model by Mann (1994)) and a definition of the *large-scale* eddies, or by means of LES simulations. Machefaux et al. (2015) showed that inconsistencies between a Mann-based and LES-based meandering process can arise due to differences in the input turbulence fields. Larsen et al. (2008) and Trujillo et al. (2011) defined the *large-scale* eddies in the order of two rotor diameters or larger as responsible for wake meandering, whereas other studies defined scales larger than 3–4D as dominant (Espana et al., 2011; Muller et al., 2015; Yang and Sotiropoulos, 2019). Albeit the severe impact of the wake meandering dynamics on load predictions, its uncertainty has not been assessed in load validation studies due to lack of data (Larsen et al., 2013; Churchfield et al., 2015; Reinwardt et al., 2018). However, aeroelastic simulations with constrained wake meandering dynamics can potentially decrease the uncertainty in load predictions under wake conditions (Conti et al., 2020c).

Further, the added turbulence formulation in the DWM model accounts for additional mechanically generated turbulence caused by the wake shear and the breakdown of tip and root vortices. These contributions are modelled by a semi-empirical formulation that uses parameters, which were calibrated against CFD simulations (Madsen et al., 2010). To our knowledge, no further development has been made on this subject.





## 1.2  Problem statement

As described above, there is no consensus for the values of the DWM model parameters when studying load predictions at any
given site. Also, and perhaps most importantly, we do not know the sources of uncertainty observed in previous studies that used
the model (Larsen et al., 2013; Churchfield et al., 2015; Reinwardt et al., 2018), which need to be addressed to provide reliable
load predictions. The common practice has been to derive optimized sets of model parameters based on limited synthetic or
experimental data. This has lead to an unknown confidence in the overall model prediction ability; incorrect calibration of the
model parameters may impact significantly the model performance and lead to suboptimal wind turbine designs. To address
this issue, we estimate uncertainties in the calibration parameters of the DWM model by applying Bayesian inference (Box and
Tiao, 1973), which consists in updating any related prior information on model parameters by incorporating new knowledge
obtained from wake flow characteristics derived through lidar measurements. Further, the Bayesian calibration provides a
systematic approach to include various types of uncertainty such as physical variability as well as measurement and modeling
errors. This paper is the first part of a two-part study dedicated on improving and validating the calibration of DWM model
parameters using lidar-derived data. This first part has a four-fold primary purpose:

1. Derive wake flow features such as the two-dimensional velocity deficit and *wake-added turbulence* profiles, as well as
   time series of the wake meandering in both lateral and vertical directions from the SpinnerLidar measurements under
   different inflow wind speeds and atmospheric stability conditions.

2. Calibrate the DWM model-based wake deficit and *wake-added turbulence* predictions using the SpinnerLidar-derived
wake flow features and the Bayesian inference framework.

3. Propagate modeling uncertainties in fully-resolved wake flow fields for robust predictions that take into account the
   calibrated uncertainties.

4. Conduct a sensitivity analysis to determine the most significant sources of uncertainty in simulated wake fields that are
   typically inputs to aeroelastic simulations.

This study contributes to the ongoing discussion regarding the accuracy of power and load predictions of wind turbines
operating under wake situations (Conti et al., 2020c,b), by quantifying uncertainties in wake simulations performed with the
DWM model under a variety of inflow wind conditions. The outcomes of this study are useful for improving currently adopted
wake simulation procedures for load analysis in the IEC standards, as well as to provide practical recommendations for wake
model calibration studies based on measurements from nacelle-mounted lidars.
The work is organized as follows. Section 2 describes the DWM model. The SWiFT layout and relative wind site conditions
are described in Sect. 3. In Sect. 4, we present the wind field retrieval assumptions used to derive wake features from Spin-
nerLidar measurements. The Bayesian calibration of the DWM model is performed in Sect. 5. We carry out the validation of
the wind turbine wake simulations and conduct a sensitivity analysis to investigate the most influential parameters in Sect. 6.
Finally, the last two sections are dedicated to the discussions and conclusions.



## 2 Dynamic Wake Meandering model


The DWM model resolves three main wake features: the quasi-steady velocity deficit, the *wake-added turbulence* and the wake meandering. Each model component is described separately in the following subsections.

### 2.1 Quasi-steady velocity deficit

The quasi-steady velocity deficit component describes the wake expansion and recovery caused partly by the recovery of the
rotor pressure field and partly by turbulence diffusion moving farther downstream of the rotor (Larsen et al., 2013). The wake deficit is formulated in the meandering frame of reference (MFoR), which is a coordinate system with origin in the center of symmetry of the deficit.

In the *far-wake* region, i.e., distances larger than two rotor diameters (Sanderse, 2015), the deficit evolution is assumed to be governed by turbulent mixing and is described by the thin shear layer approximation of the rotational symmetric N–S
equations with the pressure term disregarded (Madsen et al., 2010). To account for the neglected pressure gradient effects, an initial wake deficit is analytically formulated based on the turbine's axial induction derived from blade element momentum (BEM) theory (Madsen et al., 2010). The turbulence closure of the N-S equations is obtained by means of an eddy viscosity term, and the momentum equation is solved numerically using a finite difference scheme with the artificial initial deficit as boundary condition (Madsen et al., 2010). Here, we use the numerical scheme of the standalone DWM model (Liew et al.,
2020; Larsen et al., 2020). We refer to the generalized definition of the non-dimensional eddy viscosity term by Keck et al. (2012), who considered two major drivers to the turbulence mixing: the ambient turbulence ($TI_{amb}$) and turbulence induced by the wake shear layer:

$$\frac{\nu_T}{U_{amb}R}(r,\tilde{x}) = F_1(\tilde{x})k_1 TI_{amb} + F_2(\tilde{x})k_2 \max\left(\frac{R_w(\tilde{x})^2}{U_{amb}R}\left|\frac{\partial U(\tilde{x},r)}{\partial r}\right|; \frac{R_w(\tilde{x})}{R}\left(1-\frac{U_{min}}{U_{amb}}\right)\right),\tag{1}$$

where $\nu_T$ is the eddy viscosity, $U_{amb}$ is the ambient wind speed at hub height, and $R$ is the rotor radius. The first term to the
right-hand-side of Eq. (1) describes the contribution of the ambient turbulence and the second the self-generated turbulence by the wake shear layer. Madsen et al. (2010) proposed the instantaneous wake radius $R_w(\tilde{x})$, where $\tilde{x}$ is the downstream distance normalized by $R$, and the maximum velocity difference ($U_{amb} - U_{min}$), where $U_{min}$ is the minimum wind speed in the wake, as the turbulent length and velocity scales that govern turbulent mixing due to the wake shear layer.

Based on classical mixing length theory, Keck et al. (2012) defined the turbulence stresses to be proportional to the local
velocity gradient $\partial U(\tilde{x},r)/\partial r$, which provides a two-dimensional eddy viscosity formulation that is function of the axial and radial coordinates, $\tilde{x}$ and $r$, respectively. The max operator is included to avoid underestimating the turbulent stresses at locations where the velocity gradient of the deficit approaches zero. Both terms in Eq. (1) include a filter function ($F_1(\tilde{x})$ and $F_2(\tilde{x})$) and a model constant ($k_1$ and $k_2$). The filter functions are required to model the turbulence development behind the rotor and have values in the range 0–1 depending on the downstream distance only (Keck et al., 2012). $F_1$ accounts for the delay
of the ambient turbulence entrainment into the wake and is assumed to 'activate' ambient turbulence effects at downstream distances where the pressure has recovered ($\approx 2D$, where $D$ is the rotor diameter (Sanderse, 2015)). $F_2$ compensates for the



initial non-equilibrium between the mean velocity field and the turbulent energy content created due to the rapid change in mean flow gradients close to the rotor. We refer to Eqs. (17) and (18) in Keck et al. (2015) for the mathematical formulation of $F_1$ and $F_2$. $k_1$ and $k_2$ are calibration parameters that govern the turbulence mixing and presumably do not change with wind
turbine design and ambient conditions.

## 2.2 Wake turbulence

The wake turbulence is composed of three turbulence sources and can be defined as (Vermeer et al., 2003):

$$TI_{wake} = \sqrt{TI_{amb}^2 + TI_m^2 + TI_{add}^2}, \qquad (2)$$

where $TI_m$ denotes the turbulence induced by the meandering of the wake deficit and $TI_{add}$ is the *wake-added turbulence*. $TI_m$
is commonly denoted as the *apparent* turbulence (Madsen et al., 2005), as the stochastic meandering of the wake deficit induces additional velocity fluctuations into time series taken at fixed locations in the wake. This term is considered the main source of added turbulence in the far-wake (Madsen et al., 2010), while its spatial distribution can be computed by the convolution of the wake deficit in the MFoR, and the probability distribution function (PDF) of the wake meandering in the lateral and vertical directions (Keck et al., 2014a). $TI_{add}$ accounts for the shear- and mechanical-generated turbulence due to blade tip and root
trailing vortices. The inhomogeneity of the *wake-added turbulence* is modeled by scaling the local turbulence using the factor $k_{mt}$ (Madsen et al., 2010) as:

$$k_{mt}(r) = \mid 1 - U_{def,MFoR}(r) \mid k_{m1} + \left| \frac{\partial U_{def,MFoR}(r)}{\partial r} \right| k_{m2}, \qquad (3)$$

where $U_{def,MFoR}$ is the velocity deficit in the MFoR, and $k_{m1}$ and $k_{m2}$ are constants calibrated based on CFD results (Madsen et al., 2010). The *wake-added turbulence* derived from Eq. (3) is presumed to meander together with the wake deficit, thus being
displaced by the *large-scale* eddies in the atmosphere.

## 2.3 Meandering model

Here, the meandering model is confined to a single wake scenario, whereas multiple wake dynamics are described in Machefaux (2015). Wakes are considered to act as passive tracers driven by the *large-scale* atmospheric turbulence structures. The wake field is modeled by considering a cascade of consecutive wake deficits that are displaced by the *large-scale* lateral and vertical
velocity fluctuations, i.e., the wake transport velocities ($v_c$ and $w_c$), corresponding to the lateral ($y$) and the vertical axis ($z$), respectively. Adopting Taylor's hypothesis, the downstream advection of these deficits is assumed to be controlled by the mean wind speed of the ambient wind field. Larsen et al. (2008) estimated $v_c$ and $w_c$ by low-pass filtering atmospheric turbulence fluctuations. They defined a filtering cut-off frequency $f_{cut,off} = U_{amb}/(2D)$ thus excluding contributions from smaller eddies to the meandering dynamics. This assumption was verified using full scale lidar-based measurements collected
behind an operating turbine (Bingöl et al., 2010). The wake displacements are computed as:

$$y(x,\bar{t}) = v_c(\bar{t})\bar{t}(x) + h_{yaw}(x,\bar{t})$$
$$z(x,\bar{t}) = w_c(\bar{t})\bar{t}(x) + h_{tilt}(x,\bar{t}), \qquad (4)$$



where $\bar{t} = x/U_{amb}$ defines the time for an air particle to move from the rotor to the downstream distance in the wake region. An appropriate choice of the transport velocity of the wake advection lies between the ambient wind speed and the centre velocity of the wake deficit (Keck et al., 2014b; Machefaux et al., 2015). The contribution from the yaw misalignment, which can redirect wakes in the lateral direction, is accounted for by $h_{yaw}(x, \bar{t}) = x \tan(\theta(\bar{t}))$, where $\theta(\bar{t})$ is the yaw offset at the specific time (Machefaux, 2015; Vollmer et al., 2016). The contribution of the rotor tilt is considered by $h_{tilt}(x, \bar{t})$.

## 3 The SWiFT facility

The SWiFT facility is a research site located in Lubbock, Texas, operated by Sandia National Laboratories (Herges et al., 2017). The site includes three Vestas V27 wind turbines, two meteorological towers, and a SpinnerLidar (Peña et al., 2018) mounted on the nacelle of one of the turbines and looking backwards. The entire site is on a fiber optic data acquisition and control network that synchronizes recordings from masts, turbines, and the SpinnerLidar (Herges et al., 2017, 2018). The measurement campaign took place between 2016 and 2017 with the main objective of characterizing wake fields and investigating wake steering control strategies (Herges et al., 2017).

Figure 1 provides an overview of the test layout together with the notation used along the manuscript. In this study, we analyze data collected at the meteorological mast (*METa1*), the turbine (*WTGa1*), and the SpinnerLidar mounted on the nacelle of the *WTGa1*. The *METa1* (hereafter referred as the mast) is 60-m tall and instrumented with sonic anemometers at 10, 18, 32, 45, and 58 m, sampling at 100 Hz. Other instruments installed on the mast are reported in Herges et al. (2017). The mast is placed 2.5 D south of *WTGa1* in compliance with the IEC standard guidelines (IEC, 2015, 2017). As southerly winds are prevalent at the site (see Fig. 1-right), this layout allows to retrieve concurrent incoming wind conditions from *METa1*, wake measurements behind the *WTGa1* performed by the SpinnerLidar, and power and load measurements on the waked-*WTGa2* installed 5D downstream. The *WTGa1* and *WTGa2* are variable-speed and pitch-regulated turbines with hub height of 32.1 m, D = 27 m, and a maximum power output of 192 kW (Herges et al., 2018). The supervisory control and data acquisition (SCADA) is available for both turbines providing records of the rotor speed, pitch and yaw angles, and power production, among others, at 50 Hz.

### 3.1 SpinnerLidar

The SpinnerLidar is a research Doppler wind lidar developed at DTU based on a continuous-wave (CW) laser system (Peña et al., 2018). Hereafter, SpinnerLidar and lidar denote the same system. The SpinnerLidar has been mounted either in the spinner or on top of the nacelle of a wind turbine (Angelou and Sjöholm, 2015; Peña et al., 2018). The SpinnerLidar scans the rotor wake at high temporal and spatial resolution so that wake features can be derived. For the SWiFT campaign, the SpinnerLidar scanned continuously in a rose-pattern every 2 s (see Fig. 2), and the system internally subdivided the rose into 984 sections. The accumulated Doppler-shifted spectra at each of the sections was also recorded (Herges et al., 2017).

Once a scan was completed, the SpinnerLidar refocused at a different range and this process took about 2 s (Herges et al., 2018). For the SWiFT campaign, few scanning strategies were adopted that are described below:




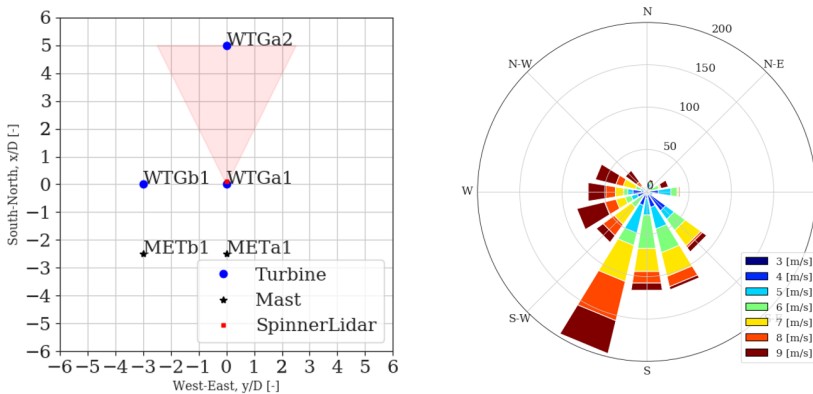

**Figure 1.** (Left): A sketch of the SWiFT layout that includes locations of the main devices (i.e., wind turbines, masts and the SpinnerLidar). The red shaded area indicates that the SpinnerLidar scans in the wake of *WTGa1* assuming winds from the south. The distances are normalized with the rotor diameter D. (Right): The wind rose at the site derived from the 32-m sonic observations collected on *METa1* during the campaign.

- *Strategy I*: the SpinnerLidar scanned seven downstream distances: 1, 1.5, 2, 2.5, 3, 4, and 5 D. A full cycle (i.e., from 1 to 5 D) took 30–42 s. This dataset is suitable for investigating the wake deficit evolution and recovery behind the rotor; however, the frequency is too low to properly derive turbulence estimates or meandering dynamics.

- *Strategy II*: the SpinnerLidar scanned at the fixed distance of 2.5 D ensuring both high spatial and temporal resolution. ≈ 298 rosette scans were generated within a 10-min period. This dataset is suited for turbulence and meandering investigations.

- *Strategy III*: the SpinnerLidar scanned at the fixed distance of 5 D behind the rotor, generating about 298 scans each 10-min. During this period, power and load measurements were recorded on *WTGa2*. This dataset is suitable for load validation analysis. Since it provides a description of the wake flow field, including velocity deficits, turbulence and meandering at a distance that corresponds to typical spacings in wind farms, it is a valuable dataset for validating fully-resolved wake flow predictions as long as induction effects are accounted for.

## 3.2  Site conditions

For extended periods of the campaign, *WTGa1* operated under large yaw misalignment, as wake steering strategies were being investigated (Herges et al., 2017). To consider periods where *WTGa1* is nearly aligned with the mean inflow, we filtered out 10-min periods characterized by an average yaw offset larger than ±10° compared to the free-stream wind direction (Conti et al., 2020a). Further, we focus the analysis on periods for which the free-stream wind direction is within 90°–270° (thus South winds, see Fig. 1-right). This leads to about 850 available 10-min periods.



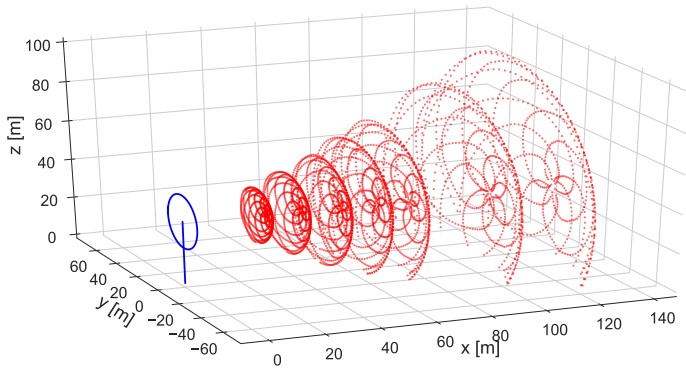

**Figure 2.** A schematic view of the SpinnerLidar's scanning patterns at several distances (i.e., 1, 1.5, 2, 2.5, 3, 4, and 5 D) behind the *WTGa1*, which is depicted in solid blue lines.

Figure 3 shows 10-min statistics of the hub-height turbulence intensity ($TI_{amb}$), the power-law shear exponent ($\alpha$), and the power production of *WTGa1* as function of the hub-height mean wind speed ($U_{amb}$) based on the mast inflow measurements. $\alpha$ is computed from the sonic measurements at 18 and 45 m. As shown, the site is characterized by a wide range of turbulence and shear conditions, which are consequence of the varying atmospheric stability (Doubrawa et al., 2019; Conti et al., 2020a).

Further, relatively low wind speeds are recorded (3–10 m/s); thus *WTGa1* operates below rated power as seen in Fig. 3 **(c)**. Because of this range of operating conditions, high rotor thrust coefficients that induce strong wake deficits characterize this dataset.

### 3.2.1 Atmospheric stability

Here, we investigate the variability of the wake flow characteristics under varying stability and inflow wind speed conditions.

We classify each 10-min sonic-derived statistic into atmospheric stability classes defined by ranges of the dimensionless stability parameter ($z/L$), where $L$ is the Obukhov length (Monin and Obukhov, 1954) computed from the sonic measurements as:

$$L = -\frac{u_*^3 \Theta}{k g \overline{w' \Theta_v'}}, \tag{5}$$

where $u_*$ is the friction velocity, $k = 0.4$ is the von Kármán constant, $g$ is the acceleration due to gravity, $T$ is the mean surface-250 layer temperature, the vertical velocity component is denoted by $w$, and $\Theta$ is the potential temperature (which we approximate by the sonic temperature). The prime denotes fluctuations around the mean value and the overbar is a time average. We define three main atmospheric stability classes based on $z/L$ ranges by Peña (2019): unstable ($-2 < z/L < -0.2$), near-neutral





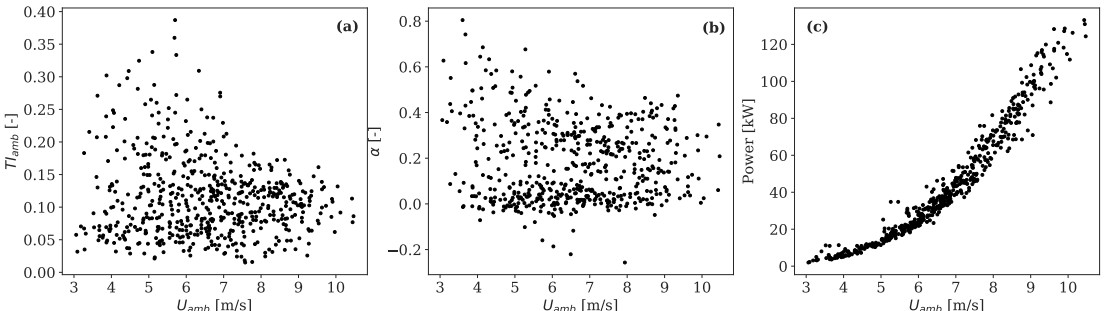

**Figure 3.** Inflow wind and operational conditions at the SWiFT site. **(a)** hub-height turbulence intensity as function of the hub-height mean wind speed based on the mast inflow measurements, **(b)** power-law shear exponent derived using observations from the 18 and 45 m sonic, and **(c)** power productions of *WTGa1* recorded from SCADA. Each marker represents a 10-min period.

($-0.2 < z/L < 0.2$), and stable ($0.2 < z/L < 2$) atmospheric conditions. We use the measurements at the 18-m sonic to derive the stability within each 10-min period.

### 3.3 Data statistics

The statistics of the inflow wind parameters are presented in separate Tables 1, 2 and 3, according to the relative SpinnerLidar scanning strategy. Table 1 presents data collected during *strategy I*. There is a sufficient amount of 10-min periods to characterize the variability of the wake deficit with respect to atmospheric stability, inflow wind speeds and downstream distances. The table shows increasing turbulence levels under unstable compared to stable cases, whereas relatively high vertical wind shears are found under stable conditions, as expected. The dataset is thus suitable for analyzing the effects of atmospheric stability on the wake recovery.

For *Strategy II*, represented in Table 2, less 10-min values are found; however this is sufficient to characterize wake turbulence and meandering under different stability conditions. For *Strategy III*, represented in Table 3, the dataset is characterized by stable conditions mainly, as the records correspond to night hours within three consecutive nights in July 2017 mainly.

### 4 Lidar measurements processing

As lidars only measure the line-of-sight (LOS) velocity ($v_{los}$), assumptions are needed to reconstruct the three-dimensional wind field $\boldsymbol{u} = (u, v, w)$, where $u$ is the longitudinal, $v$ the lateral and $w$ the vertical velocity component. If we neglect any probe volume averaging along the beam, $v_{los}$ depends on the unit directional vector $\boldsymbol{n} = (\cos\phi\cos\theta, \cos\phi\sin\theta, \sin\phi)$, which describes the scanning geometry through the elevation ($\phi$) and azimuth ($\theta$) angles, and the wind field $\boldsymbol{u}$,

$$v_{los}(\phi, \theta) = u\cos(\phi)\cos(\theta) + v\cos(\phi)\sin(\theta) + w\sin(\phi). \tag{6}$$



**Table 1.** Dataset from *Strategy I*. The data are classified according to wind speed bins of 1 m/s and three atmospheric stability classes; stable (s), near-neutral (nn) and unstable (u). The number of 10-min samples is also indicated. $\alpha$ is the power-law shear exponent; $TI_{amb}$ is the turbulence intensity defined as the standard deviation of horizontal wind speed divided by the mean wind speed. The wind speed and turbulence parameters are obtained from sonic observations at 32 m height.

| $U$ [m/s] | Samples [-] | | | $\alpha$ [-] | | | $TI_{amb}$ [%] | | |
|---|---|---|---|---|---|---|---|---|---|
| | s | nn | u | s | nn | u | s | nn | u |
| 3 | 5 | 3 | 6 | 0.39 | 0.36 | 0.08 | 7 | 10 | 18 |
| 4 | 19 | 4 | 11 | 0.30 | 0.10 | 0.01 | 8 | 19 | 22 |
| 5 | 25 | 5 | 13 | 0.27 | 0.13 | 0.01 | 7 | 11 | 22 |
| 6 | 30 | 8 | 23 | 0.28 | 0.15 | 0.04 | 7 | 11 | 16 |
| 7 | 13 | 12 | 16 | 0.23 | 0.12 | 0.02 | 7 | 12 | 13 |
| 8 | 6 | 9 | 4 | 0.27 | 0.10 | 0.04 | 7 | 12 | 10 |
| 9 | 5 | 12 | 3 | 0.30 | 0.17 | 0.02 | 7 | 11 | 9 |

**Table 2.** Similar as Table 1, but for *Strategy II*

| $U$ [m/s] | Samples [-] | | | $\alpha$ [-] | | | $TI_{amb}$ [%] | | |
|---|---|---|---|---|---|---|---|---|---|
| | s | nn | u | s | nn | u | s | nn | u |
| 5 | 2 | 4 | 12 | 0.16 | 0.07 | 0.01 | 7 | 14 | 12 |
| 6 | - | 1 | 8 | - | 0.04 | 0.01 | - | 13 | 12 |
| 7 | 9 | - | 8 | 0.22 | - | 0.10 | 10 | - | 14 |
| 8 | 3 | 5 | 1 | 0.18 | 0.12 | 0.05 | 10 | 12 | 14 |

Considering the small elevation angles and the typical low values of $w$, we assume $w = 0$ (Doubrawa et al., 2019, 2020; Debnath et al., 2019). Following the approach of Doubrawa et al. (2020), we can combine the $u$- and $v$-velocity components into a total horizontal wind vector, $U$, and Eq. (6) becomes

$$v_{los}(\phi, \theta, \bar{\theta}_0) = U \cos(\phi) \cos(\theta - \bar{\theta}_0), \tag{7}$$

where $\bar{\theta}_0$ is the yaw offset and the overbar indicates a smoothed signal, as we apply a moving average operator with a 15-s window to the yaw misalignment to account for any temporal delay from the spatial distances among the mast, turbine's nacelle and SpinnerLidar measurements (Conti et al., 2020a). With Eq. (7), we can reconstruct horizontal wind velocity measures at each individual scanned point within the rosette pattern. Further, we linearly interpolate the reconstructed wind speeds across the rosette pattern into a two-dimensional regular grid with a 2-m resolution, which is sufficient to characterize the spatial 280 characteristics of the wind field in wakes (Fuertes et al., 2018; Conti et al., 2020a).



**Table 3.** Similar as Table 1, but for *Strategy III*

| $U$ | Samples | | | $\alpha$ | | | $TI_{amb}$ | | |
| $[m/s]$ | [-] | | | [-] | | | [%] | | |
| | s | nn | u | s | nn | u | s | nn | u |
| 4 | 1 | - | - | 0.38 | - | - | 7 | - | - |
| 5 | 2 | - | - | 0.32 | - | - | 7 | - | - |
| 6 | 18 | - | - | 0.30 | - | - | 6 | - | - |
| 7 | 50 | - | - | 0.25 | - | - | 8 | - | - |
| 8 | 24 | 2 | - | 0.21 | 0.04 | - | 8 | 14 | - |
| 9 | 2 | 2 | - | 0.18 | 0.02 | - | 10 | 12 | - |

### 4.1 Lidar-estimated wake deficit

To perform comparisons with predicted velocity deficits from the DWM model, we aim at isolating the contribution of the wake deficit from that of the vertical wind shear in lidar measurements. As defined in Trujillo et al. (2011), the quasi-instantaneous wake deficit profile can be obtained by subtracting the mean vertical shear profile ($U_{amb}(z)$) from the quasi-instantaneous wake recording as:

$$U_{def}(x,y,z) = \frac{U_{amb}(z) - U(x,y,z)}{U_{amb}(z)}, \tag{8}$$

where $U(x,y,z)$ is estimated from lidar measurements using Eq. (7), and $U_{amb}(z)$ is the relative 10-min average inflow vertical wind speed profile measured at the mast. The deficit is then normalized with respect to the ambient wind speed profile. The $v_{los}$ measurements, and indeed the reconstructed $U$ wind velocities, are defined on a coordinate system that is either attached to the nacelle (nacelle frame of reference, NFoR), which rotates with the yawing of the turbine, or to the ground (fixed frame of reference, FFoR). To perform direct comparisons with the DWM model predictions, the lidar-estimated deficits obtained from Eq. (8) need to be computed in the MFoR. Here, this is performed by tracking the wake center position through the method of Trujillo et al. (2011), where a bivariate Gaussian shape is fitted to the velocity deficit flow field and the wake center is the geometric centroid of the Gaussian:

$$f_{def} = \frac{A}{2\pi\sigma_{wy}\sigma_{wz}} \exp\left[-\frac{1}{2}\left(\frac{(y_i - \mu_y)^2}{\sigma_{wy}^2} + \frac{(z_i - \mu_z)^2}{\sigma_{wz}^2}\right)\right], \tag{9}$$

where $(\mu_y, \mu_z)$ define the wake center location, $(\sigma_{wy}, \sigma_{wz})$ are width parameters of the wake profile in the $y$ and $z$ directions, respectively, $(y_i, z_i)$ denote the spatial locations of the lidar measurements, and $A$ is a scaling parameter. Each scanned point of the quasi-instantaneous wake recording can be translated into the MFoR using the estimated $\mu_y$ and $\mu_z$ from Eq. (9) (Reinwardt et al., 2020). Therefore, we can compute the multiple wake recordings within a 10-min period in the MFoR, and subsequently compute flow statistics such as the ensemble-average deficit profile as well as the spatial distribution of the wake turbulence in the MFoR. To ensure a high-quality fit, we reject scans where the estimated wake center location is within $\approx 10\%$ of the





lateral bounds of the scanning area and at more than 0.75 D from the hub height in the vertical direction (Conti et al., 2020a; Doubrawa et al., 2020).

Figure 4 illustrates ensemble-average measured deficit profiles in the MFoR at 2, 3, 4 and 5 D behind the rotor obtained
from all 10-min periods characterized by an incoming wind speed of 7 m/s, and under varying stability regimes (see Table 1 for reference). We can clearly observe the impact of the atmospheric stability and in particular of the associated turbulence levels on the wake recovery behind the rotor. A strong and well-defined symmetric wake deficit shape is seen under stable conditions (top row), whereas the deficits recover faster moving downstream as the atmosphere becomes more unstable (bottom row).

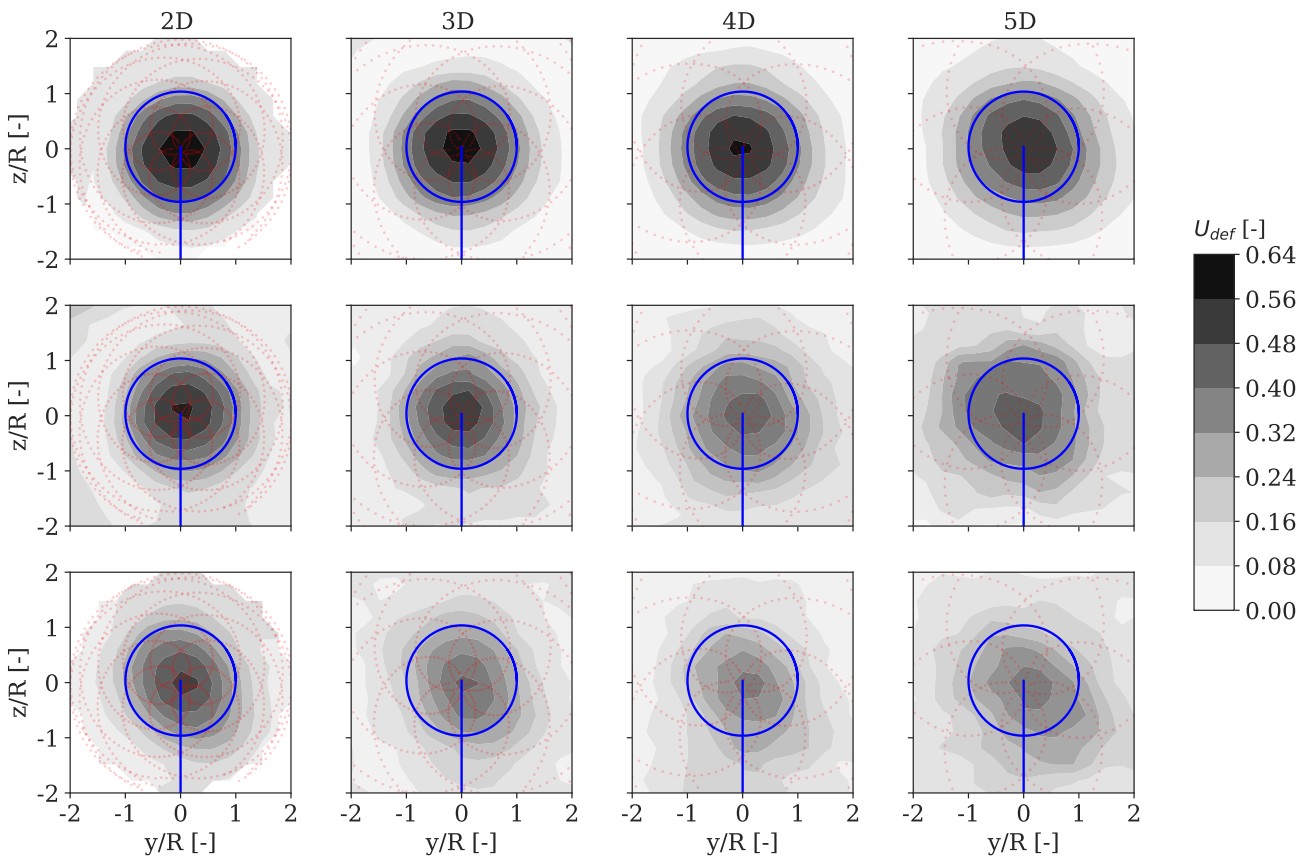

**Figure 4.** Ensemble-average velocity deficit profiles in the MFoR measured at 2, 3, 4 and 5 D behind the rotor for an inflow wind speed of 7 m/s under stable (upper row), near-neutral (middle-row) and unstable (lower row) conditions. The SpinnerLidar scanning pattern is shown in red dots, whereas the turbine rotor area is illustrated by blue solid lines. The vertical and lateral coordinates are normalized by the rotor radius and centered at hub height.





## 4.2 Lidar-estimated wake turbulence

Turbulence measures derived from lidar radial velocity measurements are 'filtered' because of their relatively large probe volume (Peña et al., 2017) and so they are generally lower than those obtained from sonic observations. Nevertheless, if the Doppler spectrum of the $v_{los}$ is available, we can potentially circumvent the averaging effects and estimate the unfiltered variance of $v_{los}$ (Peña et al., 2017; Mann et al., 2010). Mann et al. (2010) assumes that the ensemble-averaged Doppler spectrum over a time period $\langle S(v_{los}) \rangle$ is related to the probability distribution of the $v_{los}$ at the focus distance, and can be

computed as:

$$\langle S(v_{los}) \rangle = \int\limits_{-\infty}^{\infty} \varphi(s) p(v_{los}|s) ds, \tag{10}$$

where $\varphi(s)$ is the spatial averaging function of the lidar that depends on the position along the beam $s$, and $p(v_{los}|s)$ denotes the PDF of $v_{los}$ at the location $s$. If we assume that the PDF of $v_{los}$ is independent of $s$, (i.e., there is no velocity gradient along the beam), then Eq. (10) reduces to $\langle S(v_{los}) \rangle = p(v_{los})$. As a result, the $v_{los}$ statistics (i.e., mean and variance) can be

computed from the first and second central moments of $p(v_{los})$ as:

$$\mu_{v_{los}} = \int\limits_{-\infty}^{+\infty} v_{los} p(v_{los}) dv_{los}, \qquad \sigma^2_{v_{los}} = \int\limits_{-\infty}^{+\infty} (v_{los} - \mu_{v_{los}})^2 p(v_{los}) dv_{los}, \tag{11}$$

where $\mu_{v_{los}}$ and $\sigma^2_{v_{los}}$ denote the mean and unfiltered variance of $v_{los}$, respectively. Following the procedure of Peña et al. (2019), we compute the ensemble-averaged Doppler spectrum within 10-min periods by thresholding the noise-flattened spectra with a value of 1.2 and correcting them by subtracting the background spectrum. We accumulate the LOS Doppler spectra

onto the regular grid of the scanned area and estimate $\mu_{v_{los}}$ and $\sigma^2_{v_{los}}$ for each grid cell using Eq. (11). As discussed in Herges and Keyantuo (2019), invalid measurements occur due to the boresight and ground return, as well as the return from the rotating rotor of *WTGa2*, if in operation. These invalid observations appear as very high return signal in the Doppler spectrum in proximity of low wind speeds (i.e., at approximately 1 m/s) and are removed. The filtering effects due to the probe volume can be quantified by computing the ratio between filtered and unfiltered LOS variances across the rosette pattern; we find ratios in

the range 0.8–0.9 at 2.5 D, which vary according to stability conditions (not shown).

Examples of 10-min ensemble-averaged Doppler spectra obtained at three fixed locations across the scanned area, a wake center, a wake edge, and a wake-free position, are shown in Fig. 5 for an incoming wind speed of 7 m/s and low turbulence. A narrow spectrum with a single-peak distribution centered at about 7 m/s for the wake-free location (green) is seen, whereas spectrum broadening effects induced by small-scale generated turbulence are noticeable for the positions within the wake. The

335 wake center (red) shows a wider spectrum with a peak at a significantly lower wind speed than the incoming flow, whereas the wake edge (cyan) shows a double-peak distribution that may be partially due to the inhomogeneity of the wind field along the beam (Herges and Keyantuo, 2019) and also due to the meandering occurring within the analyzed 10-min period.





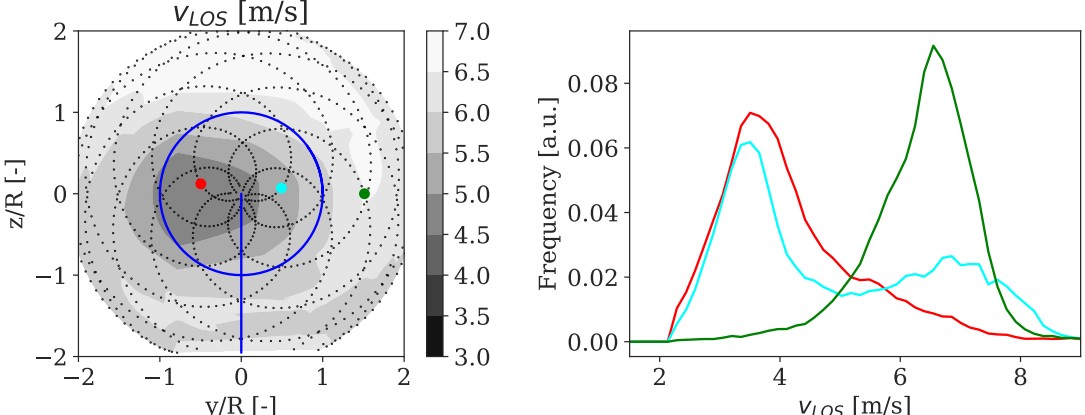

**Figure 5.** Examples of normalized Doppler LOS velocity spectra measured over a 10-min period at three different locations: wake center (red), wake edge (cyan) and wake-free (green), for an incoming wind speed of 7 m/s.

To characterize the spatial distribution of the wake turbulence within the scanned area, we derive $\sigma_U^2$ estimates directly by applying the variance operator to Eq. (7):

$$\sigma_{v_{los}}^2 = \sigma_U^2 \cos(\phi)^2 \cos(\theta - \bar{\theta}_0)^2, \tag{12}$$

where $\sigma_U^2$ is the variance of the horizontal wind speed, and as shown, covariance terms are neglected. As the LOS is almost never aligned with the $u$-velocity component across the rosette, except at the center of the pattern, $\sigma_{v_{los}}^2$ can be 'contaminated' by the variances and covariances of the other velocity components (Peña et al., 2017). Therefore, the relation in Eq. (12) can lead to inaccurate estimations of the longitudinal velocity variances. Peña et al. (2019) estimated the contamination of different components on the LOS variances for the SpinnerLidar, and showed that the ratio of the unfiltered LOS velocity variance to the variance of the longitudinal velocity component is generally lower than one across the scanned area, except at the center where the ratio is one, and within an area above the center where it can be higher than unity. Although the adopted retrieval assumption in Eq. (12) introduces uncertainties in the turbulence measures, we can account for the expected errors in the Bayesian inference framework.

Figure 6 illustrates the spatial distribution of the unfiltered $\sigma_U^2$ computed in the MFoR, normalized with the $u$-velocity variance of the ambient wind field measured at the 32-m sonic ($\sigma_{u,amb}^2$). Under stable conditions, and for a downstream distance of 2.5 D, we can observe an enhancement in turbulence levels in proximity of the rotor tips, especially in the upper part of the rotor (see Fig. 6 **(a)**). The observed added turbulence is caused by the breakdown of the rotor tip vortices. These features are no longer noticed as the atmosphere becomes more unstable, where a more uniform and less prominent distribution of the turbulence is found (see Fig. 6 **(b)** and **(c)**).



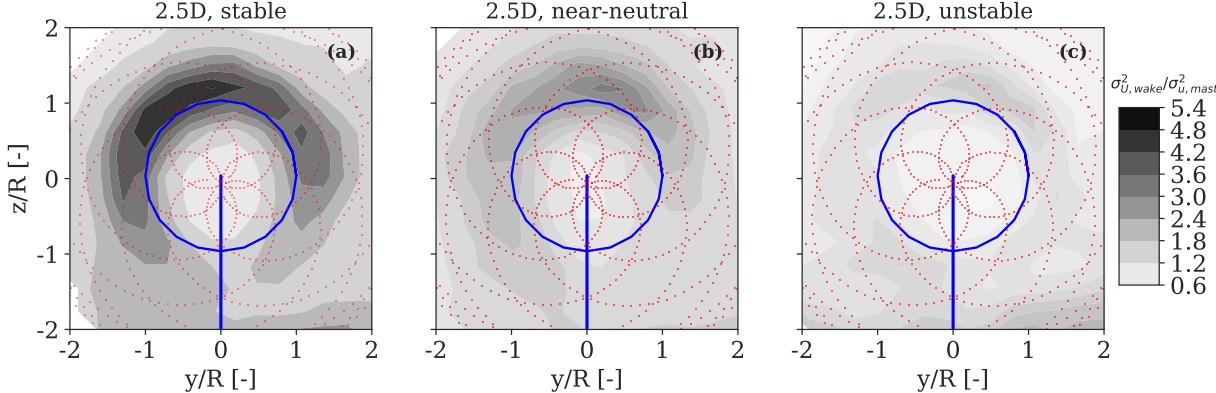

**Figure 6.** Two-dimensional spatial distribution of the horizontal wind velocity variance $\sigma_U^2$ computed in the MFoR and normalized with the $u$-velocity variance of the ambient wind field ($\sigma_{u,amb}^2$) for an incoming wind speed between 7 and 8 m/s under different stability conditions.

## 5 Calibration of the DWM model

The calibration of the wake deficit and *wake-added turbulence* components are conducted in the MFoR using a Bayesian inference framework. We describe the Bayesian model in Sect. 5.1, provide calibration results for the wake deficit in Sect. 5.2, and for the *wake-added turbulence* in Sect. 5.3. We investigate wake meandering dynamics separately in Sect. 5.4.

### 5.1 Bayesian inference formulation

The basis of the Bayesian inference is to estimate the probability distribution of the model parameters based on available observations. Let $\theta_m = \{k_1, k_2, ..., k_{m1}, k_{m2}\}$ be a set of model parameters to be estimated using lidar-derived wake features (i.e., wake deficit and *wake-added turbulence* profiles in the MFoR) denoted by $y_d = \{y_{d1}, y_{d2}, ..., y_{dn}\}$, where $n$ is the number of available observations. We consider that the experimental data and the model predictions satisfy the prediction error equation:

$$y_d = \hat{g}(\theta_m, X_m) + \epsilon, \tag{13}$$

where $\hat{g}(\theta_m, X_m)$ denotes the DWM model predictions obtained from a particular set of model parameters ($\theta_m$) and a set of observable variables ($X_m$). Here $X_m = \{TI_{amb}, U_{amb}, \alpha, C_T, x, y, z\}$ includes the inflow wind conditions measured at the mast ($TI_{amb}, U_{amb}, \alpha$), the rotor thrust coefficient of the turbine ($C_T$), which is derived from the BEM model (Madsen et al., 2010), and the spatial locations of the scanning pattern ($x, y, z$). $\epsilon = \epsilon_y + \epsilon_m$ denotes a random prediction error composed
of two terms: the measurement error $\epsilon_y$ and the model prediction error $\epsilon_m$. The former is described by a zero mean normal distribution with standard deviation $\sigma_{\epsilon_y}$, which is determined from field observations. The latter is assumed to have zero mean, which implies unbiased model predictions, and a standard deviation $\sigma_{\epsilon_m}$ to be determined by the Bayesian estimation along with the model parameters. To facilitate statistical inference, we assume $X_m$ as deterministic inputs (i.e., free of uncertainty),



and that the model error $\epsilon_m$ is independent on the set of input variables $X_m$, and described by a normal distribution. This implies that the model predictions are normally distributed for a given $X_m$, which is a reasonable choice for wake deficit profiles. The Bayesian approach for model calibration deals with updating the combined parameter set $(\theta_m, \sigma_{\epsilon_m})$, given a set of observations $(y_d, X_m)$ by applying the Bayes theorem:

$$f(\theta_m|y_d) = \frac{f(y_d|\theta_m, \sigma_{\epsilon_m})f(\theta_m, \sigma_{\epsilon_m})}{f(y_d)}, \tag{14}$$

where $f(\theta_m|y_d)$ is the updated posterior distribution of the model parameters, $f(\theta_m, \sigma_{\epsilon_m})$ is the prior distribution that is typically assigned based on subjective or previous information, $f(y_d|\theta_m, \sigma_{\epsilon_m})$ denotes the likelihood of observing the data $y_d$ from a model with corresponding $\theta_m$ parameters, and $f(y_d)$ is the prior predictive distribution that is defined as the marginal distribution $f(y_d) = \int f(y_d|\theta_m, \epsilon_m)f(\theta_m, \sigma_{\epsilon_m})d\theta_m d\epsilon_m$. By using the prediction error in Eq. (13) and assuming that the error terms are jointly normal with a zero mean vector and covariance matrix $\sum^{\epsilon_{y_d}} = diag(\sigma^2_{\epsilon_{y_d}})$ and $\sum^{\epsilon_m} = diag(\sigma^2_{\epsilon_m})$, the measured quantities follow the normal distribution $y_d \sim \mathcal{N}(\hat{g}(\theta_m, X_m|y_d), \sum_\epsilon)$, where the covariance matrix takes the form $\sum_\epsilon = \sum^{\epsilon_{y_d}} + \sum^{\epsilon_m}$. As a result, the likelihood function of observing the data follows the multi-variable normal distribution defined as:

$$f(y_d|\theta_m, \epsilon_m) = \frac{|\sum_\epsilon|^{-1/2}}{(2\pi)^{1/2}} \exp\left[ -\frac{1}{2}[y_d - \hat{g}(\theta_m, X_m|y_d)]^T \sum_\epsilon^{-1}[y_d - \hat{g}(\theta_m, X_m|y_d)] \right], \tag{15}$$

where the $|.|$ denotes the determinant. The analytical and differentiable solution of the posterior distribution of the parameters in the N-S equations with the eddy viscosity term of Eq. (1) is not readily available. Therefore, we employ a numerical sampling method to approximately evaluate the posterior distribution, and its first and second moments. Here, the adaptive No-U-Turn Markov chain Monte Carlo (MCMC) sampler is employed to generate samples from the posterior distribution (Hoffman and Gelman, 2014; Salvatier et al., 2016).

The outcome of the calibration is a joint probability distribution of the inferred model parameters. From this joint PDF, we can estimate the posterior PDF of any wake feature simulated by the DWM model, i.e., the wake deficit/*wake-added turbulence* profiles in the MFoR, or the fully-resolved wakes in the FFoR, among others, which we denote by $q$:

$$f(q|y_d) = \int_\Theta f(q|\theta_m)f(\theta_m|y_d)d\theta_m. \tag{16}$$

The posterior distribution of the wake feature $q$ in Eq. (16) can be solved numerically using sampling methods (e.g., Monte Carlo simulations) so its first and second moment can be estimated.

## 5.2 Wake deficit parameter estimation

We use lidar-derived wake deficit profiles in the MFoR collected during *Strategy I* and *Strategy II*, and employ the Bayesian model to infer uncertainty in the $k_1$ and $k_2$ parameters of the eddy viscosity term in Eq. (1). These parameters were found to be the most sensitive on the resulting wake deficit predictions (Keck et al., 2012). The prediction model of Eq. (13) is constructed as following. The experimental data $(y_d)$ comprises two-dimensional ensemble-average lidar-estimated deficit profiles binned





according to downstream distances (2, 3, 4 and 5 D), atmospheric stability (i.e., stable, near-neutral and unstable), and wind

speed bins of 1 m/s in the range 3–9 m/s. The set of observable variables comprises $X_m = \{TI_{amb}, U_{amb}, \alpha, C_T, x, y, z\}$, where the inflow parameters $(TI_{amb}, U_{amb}, \alpha)$ are provided in Tables 1 and 2; the rotor thrust coefficient $C_T$ is derived from the BEM model implemented in the aeroelastic code HAWC2 (Larsen and Hansen, 2007) and based on the aerodynamics and airfoil inputs of the SWiFT turbine (Doubrawa et al., 2020) ($C_T$=0.84 that is nearly constant for wind speeds below 9 m/s); and $x, y$ and $z$ refer to the spatial coordinates of the deficits resolved in the MFoR. The uncertainties in measured deficit profiles

are computed as $\epsilon_{y_d}(r) = \sigma(r)/\sqrt{n}$, where $\sigma(r)$ is the standard deviation of all 10-min deficits within the analyzed case at the radial position $r$, and $n$ is the number of 10-min periods (also referred to as samples in Table 1).

We select uniform prior distributions on model parameters $k_{1,prior} \sim \mathcal{U}(0.001, 0.2)$ and $k_{2,prior} \sim \mathcal{U}(0.001, 0.2)$ on intervals that consider physical constraints, ensuring convergence of results, and covering previous calibrations reported in the literature. Thus, we employ a MCMC algorithm to sample from the posterior PDFs of the calibration parameters using the Bayesian

framework. The inferred joint and marginal posterior distributions of the model parameters ($k_1$ and $k_2$) are shown in Fig. 7, together with point values from earlier studies (Madsen et al., 2010; Keck et al., 2012; Larsen et al., 2013; IEC, 2019; Reinwardt et al., 2020). As shown, the lidar-based wake deficits are informative and we obtain well-defined posteriors that follow a normal distribution with $k_1 \sim \mathcal{N}(0.081, 0.017)$ and $k_2 \sim \mathcal{N}(0.015, 0.003)$. The negative correlation between $k_1$ and $k_2$ seen in Fig. 7 indicates the interdependence of the physical induced-effects, as both parameters contribute to turbulence

diffusion. It is found that the posterior means of the informed parameters $k_1$ and $k_2$ differ from those recommended in the IEC standard (IEC, 2019). Generally, low $k_1$ and $k_2$ values attenuate the degree of turbulence mixing in the wake, which lead to strong velocity deficits persisting at farther downstream distances. We provide the statistical properties (mean, variance and coefficient of variation) of the inferred parameters in Table 4 and correlation measures in Table 5.

### 5.2.1 Wake deficit predictions in the MFoR

We propagate the uncertainties of $k_1$ and $k_2$ to predict wake deficits in the MFoR, and compare them with the ensemble-average lidar-derived profiles in Fig. 8. First, we observe that the lidar-estimated deficits exhibit a faster wake recovery as the atmosphere becomes more unstable compared to stable regimes. This effect is mainly caused by the enhanced turbulence mixing occurring under unstable conditions, as they are characterized by ambient turbulence levels 2–3 times higher than those of the stable cases (see Table 1). The lidar-observed maximum deficit varies between 30% and 60% within the first five

rotor diameters, depending on the inflow turbulence conditions. Similar behaviors were reported in recent lidar measurement campaigns (Iungo and Porté-Agel, 2014; Machefaux et al., 2016; Fuertes et al., 2018; Zhan et al., 2020b). The lidar-estimated deficits are approximately Gaussian under stable to near-neutral conditions, whereas the Gaussian shape is lost under more unstable conditions. This may result from errors in the wake tracking procedure due to the larger meandering amplitudes, and also due to the presence of large-scale turbulence structures in the inflow (Conti et al., 2020a). The rotor thrust is another factor

governing the variability of the wake recovery (Zhan et al., 2020b), however, its influence is secondary for the here-analyzed dataset due to the relatively low incoming wind speeds and relatively constant thrust coefficients (Conti et al., 2020a).





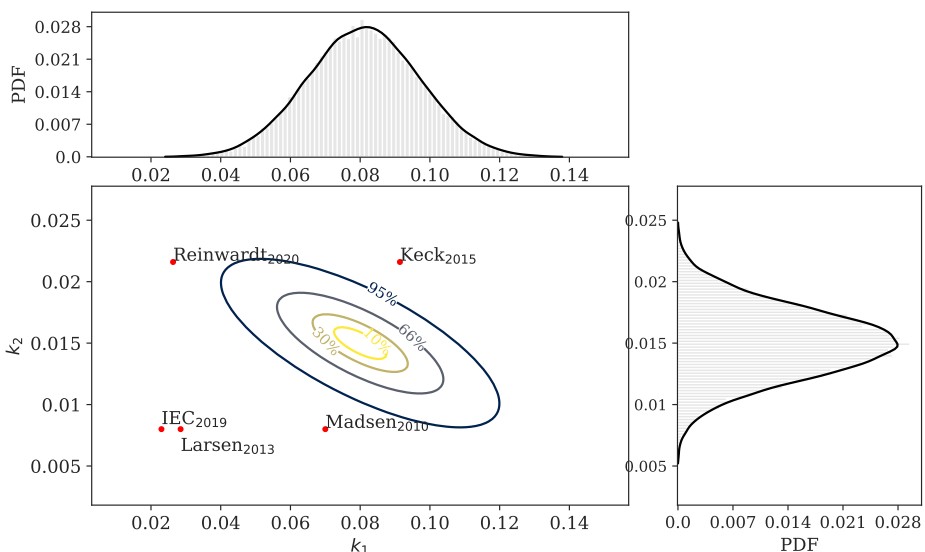

**Figure 7.** Joint and marginal posterior PDFs of $k_1$ and $k_2$ parameters. The uncertainty regions representing the 10%, 30%, 66% and 95% confidence intervals are shown. The histograms obtained from 40000 MCMC samples and the corresponding empirical PDFs are also included. The calibration parameters from early studies are shown with red markers and discussed in the text.

The DWM-predicted deficit profiles with parameters specified by their posterior distributions are in good agreement with the lidar observations for distances beyond 4D (see Fig. 8). For these distances, the turbulence mixing effects dictated by the ambient and self-generated wake turbulence on the deficit recovery are fairly well-captured by the inferred parameters. The
nominal model predictions generally fit the observations, whereas an overlap between the measurements and the region of modeling uncertainty is found. The largest deviations between predicted and measured deficits are found at shorter distances (2–3D) and mostly under unstable conditions. These deviations are mainly due to both the model inadequacy to simultaneously fit all the experimental measurements and experimental uncertainties. Similar findings were reported in Keck et al. (2015) and Machefaux et al. (2016), who showed the inaccuracy of deficit predictions at short distances.

It can be observed that the uncertainties in $k_1$ and $k_2$ parameters primarily influence the depth of the wake (i.e.,the maximum deficit), while their sensitivity decrease significantly with the outer radial distance. It is also noticed that the uncertainty of the deficit predictions increases for high ambient turbulence and far downstream distances. This is because both $k_1$ is proportional to $TI_{amb}$ in Eq. (1) and the increasing sensitivity of the model parameters to wake recovering. To provide a measure of the uncertainty of wake deficit predictions, we compute the coefficient of variation COV = $\sigma/\mu$, where $\sigma$ is the standard deviation
and $\mu$ is the mean value of the maximum deficit, obtained by propagating the PDFs of $k_1$ and $k_2$, as in Eq. (16). We find COV = 3% under stable conditions ($T_{amb}$=0.07) that increases to 6% under unstable conditions ($T_{amb}$=0.14) for an incoming wind speed of 7 m/s. This result confirms that the uncertainty of wake deficit predictions increases for higher turbulence, but it



also shows that uncertainties in $k_1$ and $k_2$ parameters do not lead to uncertainty of the same magnitude on deficit predictions resolved in the MFoR (for reference $\mathrm{COV}_{k_1} = 21\%$ and $\mathrm{COV}_{k_2} = 19\%$ as reported in Table 4).

We provide comparisons between measured and predicted wake deficit profiles using the calibration from this work as well as those reported in early studies in Fig. 9. For this particular analysis, we analyze predictions at 5D behind the rotor for an inflow wind speed of 7 m/s under stable, near-neutral and unstable regimes. The main discrepancy among the models is the relative sensitivity of the wake recovery to the ambient turbulence. This is primarily governed by $k_1$; however, in the eddy viscosity model of $\mathrm{Larsen}_{2013}$ (Larsen et al., 2013), $\mathrm{IEC}_{2019}$ (IEC, 2019), and $\mathrm{Reinwardt}_{2020}$ (Reinwardt et al., 2020), it also

depends on a nonlinear coupling function $F_{amb}(TI_{amb})$ that attenuates the wake recovery for turbulence above $\approx 12\%$ (see Fig. 6 in (Larsen et al., 2013)). This function was introduced to fit the power productions at the Egmond aan Zee offshore wind farm and it is not based on observations of the wake field (Larsen et al., 2013). The wake recovery predicted with the models of $\mathrm{Larsen}_{2013}$ (Larsen et al., 2013) and $\mathrm{IEC}_{2019}$ is practically insensitive to ambient turbulence at downstream distances up to 5D. Similar outcomes are reported in Reinwardt et al. (2020), who showed that the model of $\mathrm{Larsen}_{2013}$ provided conservative

deficits for ambient turbulence up to 16% (see Fig. 13 in Reinwardt et al. (2020)). The eddy formulation by Keck et al. (2012) in Eq. (1) is able to capture accurately the wake recovery for increasing ambient turbulence. The current calibration of the DWM model in the IEC standard provides conservative predictions especially for turbulence above 12%. Note that this can strongly impact the accuracy of power predictions within wind farms.





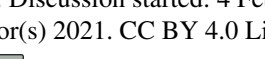

**Figure 8.** Comparison between measured and predicted ensemble-average spanwise velocity deficit profiles resolved in the MFoR at hub height and obtained at 2, 3, 4 and 5 D behind the rotor (from left to right); and for inflow wind speeds ranging 3–8 m/s with 1 m/s bin (from top to bottom panel). The SpinnerLidar-measured (markers) and DWM-predicted (solid lines) deficits are shown for each stability class (stable in blue, near-neutral in green and unstable in red). The errorbars represent the measurements uncertainty, while the shaded areas represent the uncertainty of the model predictions; both sources of uncertainty refer to the 95% confidence interval (see Table 1 for details) on the inflow wind conditions).





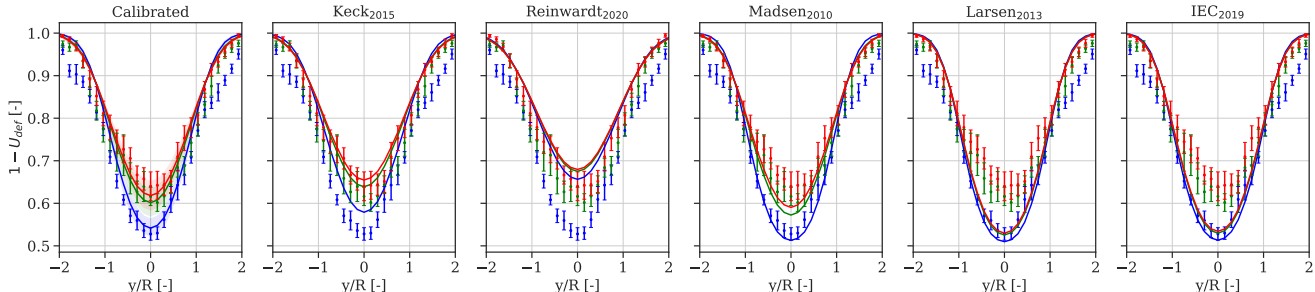

**Figure 9.** Ensemble-average spanwise velocity deficit profiles computed in the MFoR at hub height obtained at 5D behind the rotor for an incoming wind speed of 7 m/s under stable (blue), near-neutral (green) and unstable (red) atmospheric conditions. The measured turbulence intensities are 7, 12 and 16%, respectively. The SpinnerLidar-measured profiles are shown in markers and their relative 95% confidence interval by the error bars. The DWM-predicted deficits are shown in solid lines; each panel refers to model predictions using calibration parameters from a number of studies (see text for more details). The 'Calibrated' panel refers to the model proposed in the current study, while the shaded areas indicate its 95% confidence interval.

### 5.3 Improved *wake-added turbulence* formulation

The *wake-added turbulence* model (Eq. 3) assumes that turbulent structures (i.e., tip and root vortices) are unaffected by atmospheric turbulence. The rotor-induced vortices are rapidly disrupted under high turbulence conditions causing the breakdown within the first 2D (Madsen et al., 2005). However, vortices can persist and extend at farther distances under low to moderate turbulence combined with stable stratification conditions (Ivanell et al., 2009; Subramanian et al., 2018; Conti et al., 2020a). Thus, the *wake-added turbulence* profiles can exhibit a more pronounced *double-peak* feature in the proximity of the rotor tips

or a more uniform distribution depending on the atmospheric turbulence conditions (this effect is also seen in Fig. 6). Eq. (3) also assumes radially symmetric *wake-added turbulence* profiles. However, the inflow vertical wind shear re-distributes the wake turbulence. Enhanced turbulence levels are actually observed in the proximity of the upper tip of the rotor blade (Vermeer et al., 2003; Chamorro and Porte-Agel, 2009; Conti et al., 2020a). Figure 6 **(a)** shows this effect.

Due to these two assumptions, we propose an improved semi-empirical formulation of the *wake-added turbulence* scaling

factor $k_{mt}^*$, which produces wake profiles in better agreement with the lidar observations. This is achieved by relating both the depth and the velocity gradient terms to the ambient turbulence, and by including the effect of the inflow vertical wind shear on the vertical velocity deficit gradient as:

$$k_{mt}^*(y,z) = | 1 - U_{def}(y,z) | (k_{m1}^* TI_{amb} + k_{q1}^*) + \left| \frac{\partial U_{def}^*(y,z)}{\partial y \partial z} \right| (k_{m2}^* TI_{amb} + k_{q2}^*), \qquad (17)$$

where $U_{def}(y,z)^* = (U(z)U_{def}(y,z))/\max (U(z)U_{def}(y,z))$, and $k_{m1}^*, k_{q1}^*, k_{m2}^*, k_{q2}^*$ are parameters to be determined using

Bayesian inference. Figure 10 illustrates the two-dimensional profiles of the depth and velocity deficit gradient terms of Eqs. (3) and (17). As illustrated, by including the term $U_{def}(y,z)^*$, we obtain a turbulence field that mimics qualitatively well the observed enhanced turbulence within the upper wake region.




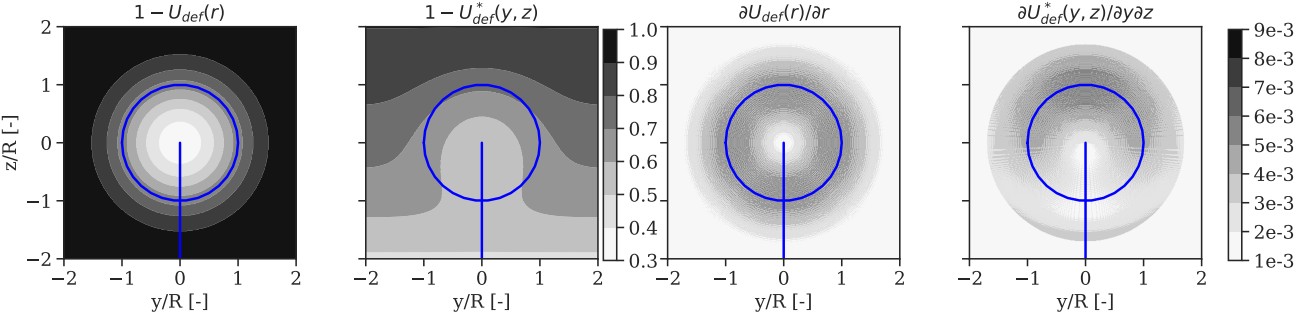

**Figure 10.** DWM-predicted flow characteristics. (Left) velocity deficit; (middle-left) velocity deficit term with combined vertical shear profile $U_{def}(y,z)^*$; (middle-right) gradient of the velocity deficit; (right) gradient of the profile resulting from the combined velocity deficit and the vertical shear. The flow characteristics are computed for $U_{amb}$ = 6 m/s, $TI_{amb}$ = 7%, and $\alpha$=0.25 at a downstream distance of 2.5 D.

### 5.3.1 Estimation of *wake-added turbulence* parameters

The calibration parameters in Eq. (17) are inferred based on the lidar-estimated *wake-added turbulence* profiles in the MFoR
collected during *Strategy II*. For this particular dataset, the SpinnerLidar scans at a fixed distance of 2.5D ensuring about 298
scans for every 10-min period. The inflow characteristics are reported in Table 2. As the Doppler LOS velocity spectrum is
available, we derive unfiltered LOS variances as described in Sect. 4.2, and subsequently the turbulence intensity as the ratio of
the standard deviation to the mean of the horizontal wind speed. From Eq. (2), we can isolate $TI_{add}$ (*wake-added turbulence*
term) by firstly resolving the wake recordings in the MFoR, which eliminates the contribution of $TI_m$, and then by subtracting
$TI_{amb}$ that is derived from the 32-m sonic observations in front of the rotor.

As a first step, we fit the 'original' analytical formulation of $k_{mt}$ from Eq. (3) to each individual lidar-estimated *wake-added
turbulence* profile to estimate the values of $k_{m1}$ and $k_{m2}$ using a simple least-squares optimization algorithm. Figure 11 **a,d**
show the relation between the estimated optimal parameters (in markers) and the ambient turbulence. It is shown that $k_{m1}$
increases and $k_{m2}$ decreases almost linearly for increasing turbulence intensity. This indicates the strong effect of the deficit
gradient term (proportional to $k_{m2}$ in Eq. (3)) under low turbulence conditions, which both amplifies the *double-peak* feature
of the wake turbulence profile at the rotor tips and mimics the wake vortices. As the ambient turbulence increases, $k_{m1}$ and
lower $k_{m2}$ become larger, which indicates a more uniform distribution of the wake turbulence. These effects are observed in
Fig. 11 **b,c** (see markers), which shows the spanwise distribution of the lidar-derived *wake-added turbulence* for two 10-min
periods with relatively low and high values of ambient turbulence intensity, 7% and 12%, respectively. We also compare the
measured and predicted vertical distribution of $TI_{add}$ in Fig. 11 **e,f**, which shows that slightly improved predictions can be
obtained using Eq. (17), i.e., considering the effects of the atmospheric shear on wake turbulence.





To infer the posterior PDFs of $k_{m1}^*, k_{q1}^*, k_{m2}^*$ and $k_{q2}^*$, we assign prior distributions based on the linear dependencies observed in Fig.11 **a,d**, namely $k_{m1,prior} \sim \mathcal{U}(0,6)$, $k_{q1,prior} \sim \mathcal{U}(0,0.1)$, $k_{m2,prior} \sim \mathcal{U}(-50,10)$ and $k_{q2,prior} \sim \mathcal{U}(0,5)$, as well as $k_1 \sim \mathcal{N}(0.081,0.017)$ and $k_2 \sim \mathcal{N}(0.015,0.003)$ previously estimated in Sect. 5.2. The uncertainties in lidar-derived *wake-*
*added turbulence* profiles account for errors introduced by the flow modeling assumptions of Eq. (12), which neglects cross-contamination effects on the LOS variance. We define these errors as zero-mean normally distributed with standard deviation $\sigma_{\epsilon_{y_d}} = 0.1 \cdot T_{add}(y,z)$, which leads to a coefficient of variation of 10% (Peña et al., 2019).

The resulting posterior PDFs of the parameters follow a normal distribution (not shown) with statistical properties tabulated in Table 4. We provide the mean values of the inferred $k_{m1}^*, k_{q1}^*, k_{m2}^*, k_{q2}^*$ values in the form of a linear regression model that
relates the depth and deficit gradient terms of Eq. (17) to the ambient turbulence in Fig. 11 **a,d**. The shaded area represents the 95% confidence interval, which is obtained by propagating the PDFs of the turbulence-related and wake deficit parameters. As shown, the predictions are characterized by a relatively high degree of uncertainty. This is because, e.g. there are few available observations to characterize turbulence, the measurements uncertainties are relatively high, and the modeling simplifications, such as the semi-empirical $k_{mt}^*$ factor. We provide the correlations among all the inferred parameters obtained from the joint
PDF in Table 5. As expected, the wake deficit parameters $k_1$ and $k_2$ are negatively correlated to $k_{m1}^*$ and $k_{q1}^*$, as $k_{m1}^*$ and $k_{q1}^*$ are proportional to the wake deficit term in Eq. (17), which is the main driver to the intensity of the *wake-added turbulence*.

## 5.4 Wake meandering

Here, we investigate the relationship between the inflow turbulence fluctuations and the lidar-tracked wake positions to characterize the *large-scale* eddies responsible for the meandering. The analysis is carried out by comparing the spectra of the
lidar-tracked meandering time series, which are derived by means of the tracking algorithm in Eq. (9), with that simulated from the meandering model in Eq. (4), where $v_c$ and $w_c$ are obtained from the sonic observations at 32 m. Yaw misalignment from SCADA is also included. We conduct the analysis using the dataset collected at 2.5 D behind the rotor during *Strategy II*, and classify all the available 10-min periods according to stability to derive ensemble-average spectra from multiple observations with similar inflow conditions. Note that the measured inflow wind speeds are below rated, and turbulence levels from all 10
min periods within each stability class are similar as those reported in Table 2.

Results of the spectral analysis for both lateral and vertical meandering are shown in Fig. 12. Here, the ensemble-average spectra from the SpinnerLidar observations are compared to those from the meandering model without low-pass filtering the incoming turbulence fluctuations (denoted as $DWM^{wf}$). The spectra are normalized with their relative variances and plotted as function of the commonly used Strouhal number $St = fD/U_\infty$, where $f$ denotes frequency and $U_\infty$ is the aggregated wind
speed from the ensemble-average statistics. As shown, the slope of the lidar-based spectra (red lines) matches that of $DWM^{wf}$ (blue lines) up to $St$ = 0.3–0.5, which corresponds to three and two rotor diameters, respectively. For $St > 0.5$, the energy content of the lidar-estimated spectra remarkably decreases compared to that of $DWM^{wf}$. These observations indicate that *large-scale* turbulent structures ($>2D$) are dominant in the wake meandering (Trujillo et al., 2011; Heisel et al., 2018). When compared to the stable case, the spectra under unstable conditions show a higher energy content at large turbulence scales, or
equivalently low Strouhal number.



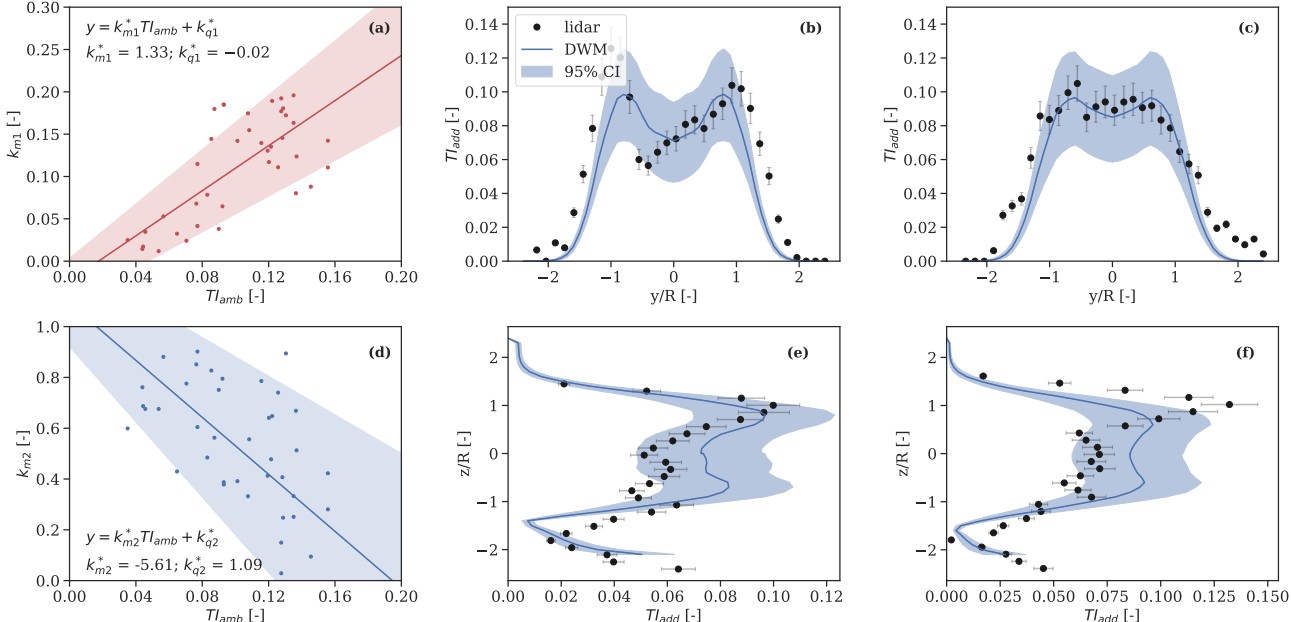

**Figure 11.** *Wake-added turbulence* predictions. **(a)** and **(d)** show the relation of both $k_{m1}$ and $k_{m2}$ in Eq. (3) to ambient turbulence. The linear regression model that is determined using Bayesian inference is shown in solid lines, whereas the shaded areas indicate the 95% confidence interval obtained by propagating the posterior PDFs of the parameters. The comparison between measured (markers) and predicted (lines) lateral *wake-added turbulence* profiles resolved in the MFoR at hub height and obtained at 2.5D behind the rotor are shown in **(b)** and **(c)**, for $TI_{amb} = 7\%$ and 12%, respectively, whereas the relative vertical profiles are shown in **(e)** and **(f)**. The error bars indicate the measurements uncertainty, whereas the shaded areas that of the model predictions relative to the 95% confidence interval.

Figure 12 shows that a stochastic description of the *large-scale* eddy size might be appropriate. Thus, we describe the *large-scale* eddies responsible for the wake meandering by introducing the stochastic variable $D_m$, which is normally distributed with mean equal to 2.5D (it corresponds to the wake diameter at 5D behind the rotor) and a standard deviation of 0.3D based on the observations in Fig. 12. The resulting 95% confidence intervals are shown as shaded areas in Fig. 12. The uncertainty in

$D_m$ is found negligible when computing wake meandering time series; this is shown in Appendix A.

## 6 Validation of the DWM model

The validation is performed by resolving wake fields in the FFoR, thus the simulated wakes include the combined effects of the velocity deficit, added turbulence, and wake meandering dynamics in both lateral and vertical directions. This analysis is carried out using data from *Strategy III*, i.e., at a fixed distance of 5D behind the rotor, ensuring a sufficient amount of scans

to derive unfiltered turbulence estimates as well as wake meandering time series within a 10-min period. The analyzed dataset



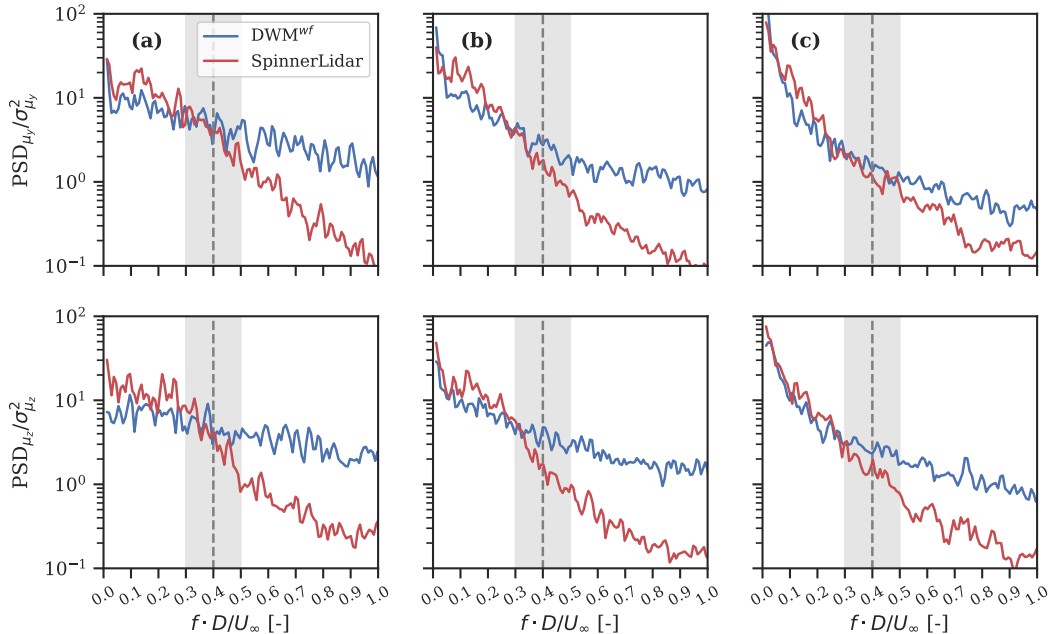

**Figure 12.** Normalized ensemble-average PSD of the lateral and vertical wake meandering tracked by the SpinnerLidar (red) and that derived using the meandering model (DWM$^{wf}$ shown in blue), under stable (column **a**), near-neutral (column **b**), and unstable conditions (column **c**). The ensemble-average PSDs are computed for data collected at 2.5 D behind the rotor and are normalized with their relative variances. The 95% confidence interval in the *large-scale* eddies definition is shown by the grey area (see text for more details).

is primarily characterized by stable conditions as seen in Table 3, i.e., low turbulence intensities (6–8%) and strong vertical shears ($\alpha = 0.18$–$0.38$). The wind speed is mainly lower than 9 m/s, thus *WTGa1* operates below rated power.

### 6.1 Correction for rotor induction effects

The SpinnerLidar measurements collected during *Strategy III* were taken in the induction zone of the *WTGa2*. For induction
correction, we employ the two-dimensional induction model of Troldborg and Meyer Forsting (2017), which accounts for both longitudinal and radial variation of the induced wind velocity:

$$C_{ind} = \left[ 1 - a_0 \left( 1 - \frac{\xi_x}{\sqrt{1 + \xi_x^2}} \right) \cdot \left( \frac{2}{\exp\left(+\beta_a \epsilon_a\right) + \exp\left(-\beta_a \epsilon_a\right)} \right)^2 \right], \tag{18}$$

where $a_0$ is the induction factor at the rotor center area, $a_0 = 0.5(1 - \sqrt{1 - \gamma_a C_T})$, $\gamma_a = 1.1$ (Troldborg and Meyer Forsting, 2017), $\xi_x = x/R$ is the distance upfront the rotor normalized by the rotor radius, $\rho_a = \sqrt{y^2 + z^2}/R$ denotes the radial dis-
tance from the rotor center axis, and $\epsilon_a = \rho_a/\sqrt{\lambda_a(\eta_a + \xi_x^2)}$ being $\beta_a = \sqrt{2}$, $\alpha_a = 8/9$, $\lambda_a = 0.587$, $\eta_a = 1.32$ (Troldborg and Meyer Forsting, 2017; Dimitrov, 2019). The lidar-measured wind speed across the scanned area is scaled by the induction



factor in Eq. (18). The estimated induction factors indicate that the wind speed can be reduced at hub height by up to 12% upstream the *WTGa2* and below rated power (not shown).

## 6.2 Uncertainty propagation of simulated wake fields in the FFoR

We derive the two-dimensional spatial distribution of the mean wind speed in the wake region ($U_{FFoR}$) by the convolution between the wake deficit in the MFoR ($U_{def,MFoR}$), and the PDF of the meandering path ($f_m$) (Keck et al., 2015):

$$U_{FFoR}(y, z, k_1, k_2, D_m, y_{m,\epsilon}, z_{m,\epsilon}) =$$
$$U_{amb}\left(\frac{z}{z_{hub}}\right)^{\alpha} \int \int U_{def,MFoR}(y - y_m + y_{m,\epsilon}, z - z_m + z_{m,\epsilon}, k_1, k_2) \cdot f_m(y_m, z_m, D_m, y_{m,\epsilon}, z_{m,\epsilon}) dy_m dz_m, \quad (19)$$

where $(y_m, z_m)$ denote the spatial coordinates of the wake meandering time series, and $(y_{m,\epsilon}, z_{m,\epsilon})$ are measures of their relative uncertainties. We introduce these errors to account for incorrect wake tracking positions that can arise due to the adopted

wake tracking algorithm. $y_{m,\epsilon}$ and $z_{m,\epsilon}$ are assumed to be uncorrelated and to follow a normal distribution with zero mean and standard deviation such as the 95% percentile corresponds to approximately 4 m, which is twice the resolution adopted to interpolate SpinnerLidar measurements onto the regular grid (see Sect. 4). Note that the atmospheric shear profile $U_{amb}(z)$ is superposed after the wake deficit calculation (Madsen et al., 2010). Similarly, the two-dimensional spatial distribution of the $u$-velocity variance ($\sigma^2_{u_{FFoR}}$) can be computed as:

$$\sigma^2_{u_{FFoR}}(y, z, k_1, k_2, k^*_{m1}, k^*_{q1}, k^*_{m2}, k^*_{q2}, D_m, y_{m,\epsilon}, z_{m,\epsilon}) = \sigma^2_{u_{amb}} +$$
$$\int \int ((U_{MFoR}(y - y_m + y_{m,\epsilon}, z - z_m + z_{m,\epsilon}, k_1, k_2) - U_{FFoR}(y - y_m + y_{m,\epsilon}, z - z_m + z_{m,\epsilon}))^2 +$$
$$\left(k^*_{mt,MFoR}(y - y_m + y_{m,\epsilon}, z - z_m + z_{m,\epsilon}, k_1, k_2, k^*_{m1}, k^*_{q1}, k^*_{m2}, k^*_{q2}) \cdot U_{MFoR}(y - y_m + y_{m,\epsilon}, z - z_m + z_{m,\epsilon}, k_1, k_2))^2\right)$$
$$\cdot f_m(y_m, z_m, D_m, y_{m,\epsilon}, z_{m,\epsilon}) dy_m dz_m,$$

$$(20)$$

where $U_{MFoR} = U_{amb}(z) U_{def,MFoR}$, $U_{FFoR}$ is derived from Eq. (19), and $\sigma^2_{u_{amb}}$ is the variance of the ambient $u-$velocity component. Alternatively, $U_{FFoR}$ and $\sigma^2_{u_{FFoR}}$ can be equivalently derived by superposing the wake deficit and the *wake-added turbulence* factor $k^*_{mt}$ on stochastic turbulence fields with constrained meandering path, and subsequently by computing the first- and second-order statistics of the synthetic wind fields. However, the analytical forms in Eqs. (19) and (20) can be easily

used to propagate the posterior PDFs of the calibration parameters to predict profiles of $U_{FFoR}$ and $\sigma_{u_{FFoR}}$ with relative uncertainties. The inflow parameters ($U_{amb}$, $\alpha$ and $\sigma^2_{u_{amb}}$) in Eqs. (19) and (20) are required inputs for the DWM model and are estimated from mast measurements.

We compute $U_{FFoR}$ and $\sigma_{u_{FFoR}}$ from Eqs. (19) and (20) by constraining the meandering path ($f_m$) either using the lidar-tracked meandering time series, which we denote as DWM*, or by using the meandering model in Eq. (4) with low-pass filtered

$v_c$- and $w_c$-velocity fluctuations obtained from the 32-m sonic and the yaw misalignment of *WTGa1* obtained from SCADA, which we denote as DWM**. In the latter model, the low-pass filtered frequency is defined as function of the stochastic



**Table 4.** Mean ($\mu$), standard deviation ($\sigma$) and coefficient of variation (COV$= \sigma/\mu$) estimated from the posterior PDFs of model parameters. The values of $k_1$, $k_2$, $\sigma_{\epsilon_{def}}$, $k_{m1}$, $k_{q1}$, $k_{m2}$, $k_{q2}$ and $\sigma_{\epsilon_{add}}$ are determined using Bayesian inference. $D_m$ denotes the spatial size of the *large-scale* eddies governing wake meandering dynamics, and $y_{m,\epsilon}$ and $z_{m,\epsilon}$ denote the wake tracking position errors expressed in meters.

| Modules | wake deficit | | | *wake-added turbulence* | | | | | wake meandering | | |
|---|---|---|---|---|---|---|---|---|---|---|---|
| | $k_1$ | $k_2$ | $\sigma_{\epsilon_{def}}$ | $k_{m1}^*$ | $k_{q1}^*$ | $k_{m2}^*$ | $k_{q2}^*$ | $\sigma_{\epsilon_{add}}$ | $D_m$ | $y_{m,\epsilon}$ [m] | $z_{m,\epsilon}$ [m] |
| $\mu$ | 0.081 | 0.015 | 0.05 | 1.33 | -0.02 | -5.61 | 1.09 | 0.03 | 2.5 | 0 | 0 |
| $\sigma$ | 0.017 | 0.003 | 0.004 | 0.14 | 0.014 | 0.97 | 0.09 | 0.001 | 0.3 | 1.5 | 1.5 |
| COV | 21% | 19% | 7% | 10% | 59% | 28% | 10% | 3% | 12% | - | - |

**Table 5.** Correlation coefficients between model parameters estimated using Bayesian inference.

| | $k_1$ | $k_2$ | $\sigma_{\epsilon_{def}}$ | $k_{m1}^*$ | $k_{q1}^*$ | $k_{m2}^*$ | $k_{q2}^*$ | $\sigma_{\epsilon_{add}}$ |
|---|---|---|---|---|---|---|---|---|
| $k_1$ | 1 | -0.73 | <0.01 | -0.15 | -0.02 | 0.02 | -0.06 | -0.06 |
| $k_2$ | -0.73 | 1 | <0.01 | -0.12 | <0.01 | <0.01 | -0.01 | <-0.01 |
| $\sigma_{\epsilon_{def}}$ | <0.01 | <0.01 | 1 | <0.01 | <0.01 | <0.01 | <0.01 | <0.01 |
| $k_{m1}^*$ | -0.15 | -0.12 | <0.01 | 1 | -0.61 | -0.21 | -0.57 | -0.01 |
| $k_{q1}^*$ | -0.02 | <0.01 | <0.01 | -0.61 | 1 | 0.19 | -0.26 | 0.06 |
| $k_{m2}^*$ | 0.02 | <0.01 | <0.01 | -0.21 | 0.19 | 1 | -0.47 | -0.03 |
| $k_{q2}^*$ | -0.06 | -0.01 | <0.01 | -0.57 | -0.26 | -0.47 | 1 | -0.03 |
| $\sigma_{\epsilon_{add}}$ | -0.06 | <0.01 | <0.01 | -0.01 | 0.06 | -0.03 | -0.03 | 1 |

variable associated to the size of the *large-scale* eddies ($D_m$) as discussed in Sect. 5.4. Figure 13 shows the comparison between observed and predicted (with both DWM* and DWM** models) profiles of the mean wind speed and its standard deviation obtained from two different 10-min periods.

A good agreement between measurements and predictions is found. The vertical profile of the mean wind speed exhibits a *single-peak* shape resulting from the combined effects of the inflow vertical shear (modeled by a power-law) and the wake-induced Gaussian-like deficit shape. The wake turbulence in the lateral direction exhibits a *double-peak* shape with larger values near the locations associated with strong velocity gradients that are further enhanced by the wake meandering. Enhanced turbulence levels in proximity of the upper wake region are observed from lidar measurements. The deviations between

DWM*- and DWM**-model predictions are more pronounced for the simulated turbulence than for the wind speed fields, and are exclusively due to differences in the meandering representations.

The wake simulations uncertainties shown in Fig. 13 are determined by propagating uncertainties in model parameters (Table 4), and by accounting for relative correlations (Table 5) using Monte Carlo simulations. The uncertainties of lidar-measured wind speeds account for volume averaging effects and for errors introduced by the retrieval assumptions (e.g., $w\sin(\phi) = 0$

m/s) (Debnath et al., 2019). The uncertainties of lidar-derived turbulence (here defined as the standard deviation of the wind speed) account for errors introduced by neglecting cross-contamination effects in Eq. (12) (Peña et al., 2019).



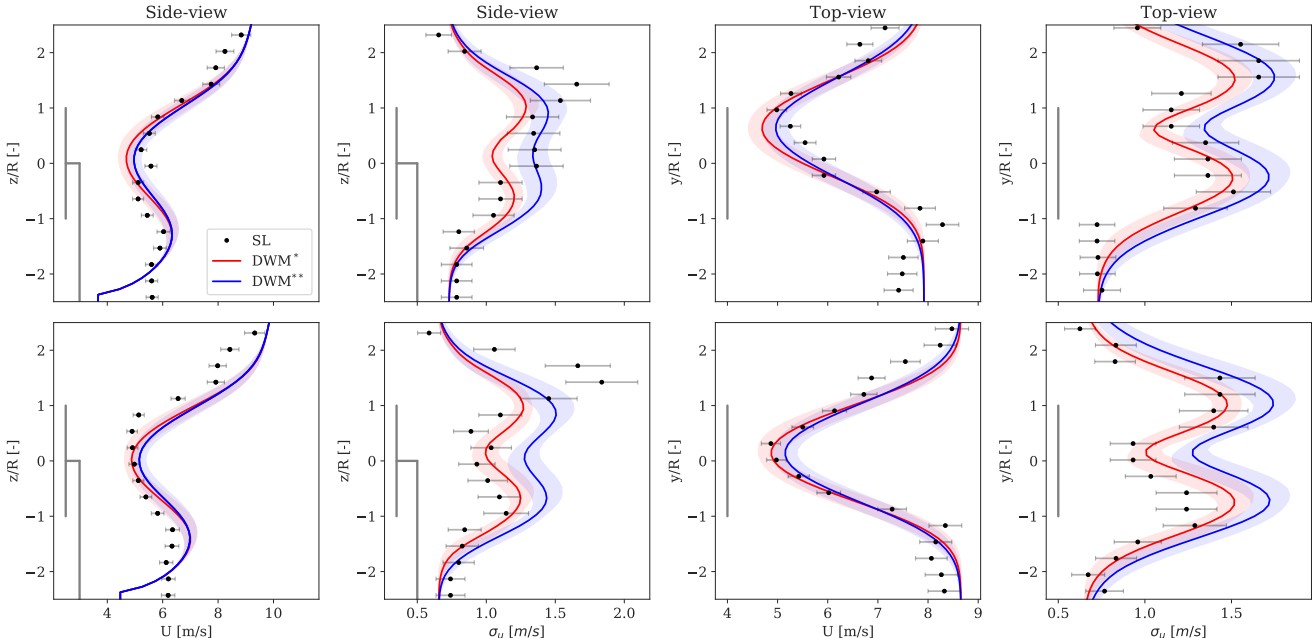

**Figure 13.** Comparisons between SpinnerLidar-measured (SL) and DWM-predicted spatial distribution of the mean and standard deviation velocity computed in the FFoR and obtained at 5D in the wake region for two 10-min periods. The predictive DWM model that incorporates the SpinnerLidar-tracked meandering time series are denoted by DWM* (red line), and that based on the meandering model complemented with measured inflow turbulence fluctuations from the mast and yaw offsets are denoted by DWM** (blue line). The solid lines represent the mean predictions, whereas the shaded areas indicate the 95% confidence intervals. The uncertainties in measurements are shown with errorbars. The top-view profiles are centered at hub height, whereas the side-view profiles along the vertical symmetry plane of the wake.

To evaluate the performance of the DWM model, we calculate two flow metrics that are relevant in aeroelastic simulations (Dimitrov et al., 2018; Murcia Leon et al., 2018) and compare them to relative measured quantities: the rotor effective wind speed ($U_{eff}$) defined as the weighted sum of wind speeds across the rotor area, and the effective wake turbulence ($\sigma_{u,eff}$) that is derived as the weighted sum of turbulence estimates across the rotor. Figure 14 shows the one-to-one comparison between measured and DWM*-model predicted flow metrics for all 10-min periods. Similar statistics are obtained with the DWM**-model (not shown). We find a slope that deviates $< 1\%$ from unity and $R^2 = 0.95$ for $U_{eff}$ and a bias of 4% and $R^2 = 0.93$ for $\sigma_{u,eff}$. The observed scatter is explained by the large measurement uncertainties, by the uncertainties of inflow wind parameters, and by those of the model predictions. The latter is estimated by propagating uncertainties of model parameters provided in Table 4, which result in a COV of 1% for $U_{eff}$, and of 3% for $\sigma_{u,eff}$. These findings indicate that the inferred uncertainties in model parameters do not propagate into an error of the same magnitude on the fully-resolved wakes that are eventually input to aeroelastic simulations.





Further, a sensitivity analysis indicates that the variations in $U_{eff}$ and $\sigma_{u,eff}$ are mostly explained by the uncertainties in $k_1$ and $k_2$ and the wake tracking position $y_{m,\epsilon}, z_{m,\epsilon}$. This is shown in Appendix B. The contribution of the *wake-added turbulence* from Eq. (17) on the total wake turbulence $\sigma_{u,eff}$ in the FFoR is marginal and accounts for approximately 2–7%. The turbulence induced by the meandering of the wake deficit is thus the major source of added turbulence (Madsen et al., 2010).

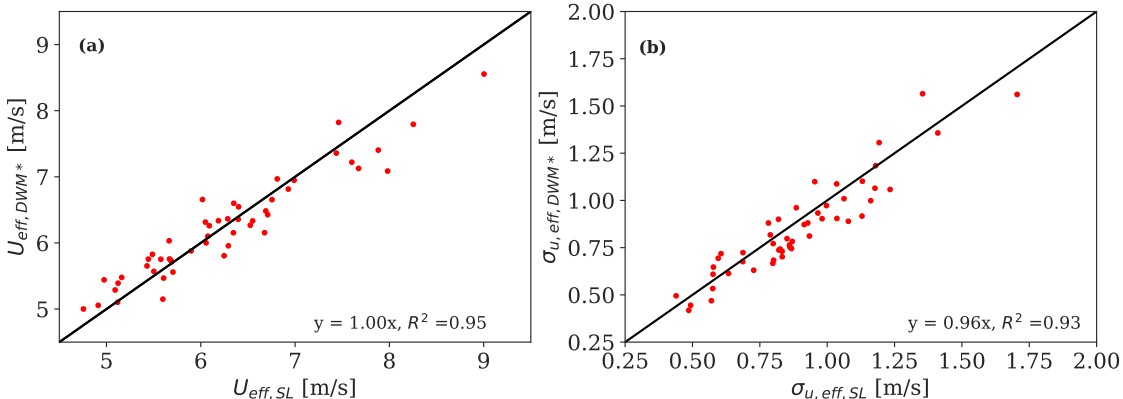

**Figure 14.** Comparison of the SpinnerLidar-measured (SL) and $\mathrm{DWM}^{*}$-predicted rotor-effective wind speeds $U_{eff}$ **(a)**, and rotor-effective turbulence $\sigma_{u,eff}$ **(b)**. The $\mathrm{DWM}^{*}$ predictions are obtained by constraining the meandering on the basis of SpinnerLidar-tracked wake displacements.

## 7 Discussion

Atmospheric stability significantly impacts wake evolution and recovery (Iungo et al., 2013; Zhan et al., 2020b). SpinnerLidar measurements of the wake show that the wake recovers faster under unstable compared to stable conditions primarily due to the high turbulence levels of the former. The wake effects can be predicted accurately by the DWM model when using appropriate calibration parameters. Further, accurate reproduction of the wake meandering dynamics in both lateral and vertical directions is key for accurate wake simulations.

We recommend conducting DWM model calibrations by resolving the wake flow features such as the wake deficit and *wake-added turbulence* profiles in the MFoR. This requires high spatial and temporal resolution scanning strategies, which cover the two-dimensional plane upfront the rotor to track both the horizontal and vertical wake displacements, and to reconstruct the spatial distribution of the wind and turbulence fields. When using power production for such calibrations (the IEC standard (IEC, 2019) values are based on this approach), we are unable to distinguish between uncertainties from inaccurate wake deficit predictions and those from erroneous wake meandering representations. The latter plays an important role in the accuracy of the fully-resolved wake fields, which are inputs to aeroelastic simulations.



Datasets that include observations of the wake deficit profiles under varying stability conditions, inflow wind speeds, and downstream distances are imperative. Here, we find that the velocity deficit's recovery rate for increasing turbulence (see Fig. 9) is the main difference among calibrations.

As wind turbines are typically spaced 5D and beyond, it is recommended to focus future measurement campaigns to include those regions. To this extent, Bayesian inference is a valuable approach for updating the PDFs of model parameters estimated within this work by directly including data from future observations while retaining information from the earlier observations. Power and load validation analyses using the proposed calibration parameters at multiple sites can further increase the confidence in our calibration methodology.

Characterizing wake turbulence using lidars is challenging due to the limited sampling frequency and probe volume effects (Peña et al., 2017). We demonstrate the usefulness of Doppler radial velocity spectra to compute unfiltered LOS variances in wake conditions. In addition to the enhanced turbulence intensity, the wake turbulence is characterized both by being highly isotropic and reduced turbulence length scale compared to the ambient turbulence (Madsen et al., 2005). These turbulence characteristics were not investigated in this work.

## 8 Conclusions

We analyzed high spatial and temporal resolution SpinnerLidar measurements of the wake field collected at the SWiFT facility and derived wake features such as the wake deficit, *wake-added turbulence*, and wake meandering under varying atmospheric stability conditions, inflow wind speeds, and downstream distances. The SpinnerLidar-estimated wake characteristics computed in the MFoR were used to determine uncertainties in the DWM model parameters using Bayesian inference. The uncertainties in model parameters were propagated to predict fully-resolved wake flow fields in the FFoR. This approach allowed us quantifying uncertainties in the DWM-simulated wake fields and to investigate the sensitivity of the DWM model parameters on flow features that primarily affect power and load predictions.

The SpinnerLidar-derived wake deficit profiles revealed the strong impact of atmospheric stability on wake evolution. In particular, we observed the faster recovery of the deficit under unstable compared to stable regimes, as higher turbulence intensities characterized the former. These effects were accurately reproduced by the eddy viscosity term of the DWM model with the inferred parameters for distances beyond 4D. Our results indicate that the currently adopted parameters in the IEC standard lead to conservative velocity deficit predictions (up to 18% for moderate to high ambient turbulence $TI_{amb} \geq 12\%$) at distances up to 5D behind the rotor.

We proposed and verified an improved semi-empirical formulation of the *wake-added turbulence* model that captured the effects of the atmospheric shear and the ambient turbulence on the spatial re-distribution of the wake turbulence observed at 2.5 D. We also demonstrated that the wake meandering is the major source of added turbulence in the wake region.

The underlying hypothesis of the DWM model, i.e., wakes are advected passively by the *large-eddies* in the incoming wind field, was verified by means of the SpinnerLidar-tracked meandering time series. The spectral analysis indicated that *large-eddies* associated with sizes larger than 2D are the main responsible for the wake meandering; however, the *large-eddies*





'definition' had only marginal effects on the predicted wake fields. Accurate tracking of the wake center position was the most
influential factor in simulating wake flow fields accurately. We expect that it also plays a central role in the accuracy of power
and load predictions.

In the *Part II* of this work, we will quantify uncertainties in power and load predictions based on the proposed calibration at
two different sites, the SWiFT facility and the Nørrekær Enge wind farm in Denmark (Peña et al., 2017; Dimitrov, 2019; Conti
et al., 2020b).


## Appendix A: Comparisons of wake meandering time series

Figure A1 compares measured and predicted time series of the wake meandering in the lateral direction under varying stability
conditions. The DWM predictions are computed by applying the filtering cut-off frequency with the stochastic definition of
the *large-scale* eddies ($D_m$). A reasonable agreement between the two signals is found, where the major wake displacements
are captured by the meandering model. Note that improved correlation can be achieved by utilizing a reduced advection wind
speed in the time-lag parameter in Eq. (4) than the ambient wind velocity. Nevertheless, the largest observed movements are
induced by the yawing of the *WTGa1*, which is fairly frequent within the analyzed dataset. Further, the 95% confidence interval
of the meandering predictions obtained by propagating uncertainties in the $D_m$ definition are nearly negligible. This indicates
that uncertainties in the filtering cut-off frequency used in the DWM formulation has a marginal effect on the accuracy of wake
simulations.

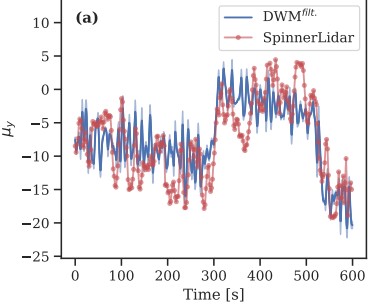
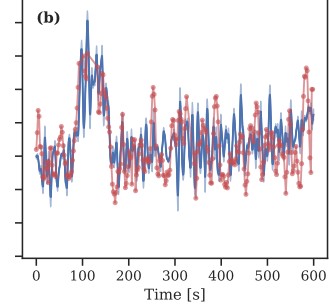
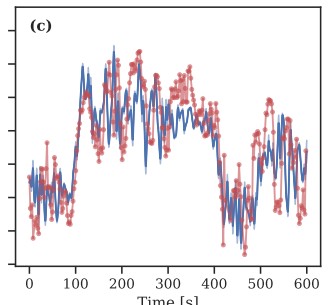

**Figure A1.** Time series of the wake meandering in the lateral direction observed by the SpinnerLidar (red markers) and those derived from the meandering model of Eq. (4) denoted as DWM$^{filt.}$ (blue lines) under stable (**a**), near-neutral (**b**), and unstable conditions (**c**). The predictions are obtained at 2.5 D behind the rotor. The blue shaded areas indicate the 95% confidence intervals obtained by propagating the uncertainty in the *large-scale* eddies definition



## Appendix B: Sobol sensitivity indices

Sensitivity analysis is conducted to identify the most important parameters affecting the accuracy of wake simulations. Here, we only investigate the uncertainties in the model parameters, which are listed in Table 4, and assume that the inflow conditions are perfectly prescribed. We employ a variance-based sensitivity method and compute total Sobol indices (Sobol, 2001; Saltelli

et al., 2010). The Sobol sensitivity decomposes the variance of the response (e.g., $U_{eff}$ and $\sigma_{U,eff}$) into contributions from input parameters and associated interactions.

The Sobol indices computed from wake simulations at five rotor diameters behind the rotor are illustrated in Fig. B1. Note that uncertainties in wake center locations ($y_{m,\epsilon}$ and $z_{m,\epsilon}$) have a similar influence as for the calibration parameters (i.e., $k_1$ and $k_2$) on the predicted flow features such as $U_{eff}$ and $\sigma_{u,eff}$. Overall, tracking the wake meandering in both lateral and vertical

directions is of primary importance in wake field representations and therefore in power and load validations. The sensitivity of the *wake-added turbulence* model parameters as well as the *large-scale* eddies definition ($D_m$) are marginal. It is inferred that these parameters can be considered as deterministic without compromising the accuracy of wake simulations.

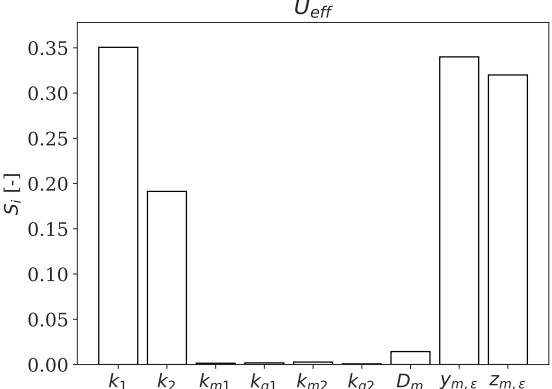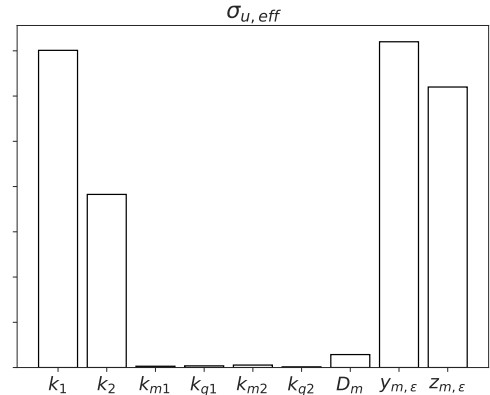

**Figure B1.** Variance decomposition (Sobol indices) for the the rotor-effective wind speed **(a)** and a measure of the rotor-effective turbulence **(b)**.



*Acknowledgements.* Sandia National Laboratories is a multi-mission laboratory managed and operated by National Technology and Engineering Solutions of Sandia, LLC., a wholly owned subsidiary of Honeywell International, Inc., for the U.S. Department of Energy's National Nuclear Security Administration under contract DE-NA-0003525.



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
