# Peer review of "Probabilistic estimation of the Dynamic Wake Meandering model parameters using SpinnerLidar-derived wake characteristics"

_Wind Energy Science, 2020_

## Author Comment (AC1)

**Author response to reviewer 1**

The authors response is shown in red

We thank the reviewer for the valuable comments and suggestions, which we consider very important and help us to sharpen and improve the manuscript. Here our response to each comment.

The manuscript describes the calibration of two of the three model components that constitute the DWM model and the comparison of the full model result to wake flow measurements. Data base for calibration are measurements from a SpinnerLidar in the wake field of one of the V27 wind turbines at the SWIFT facility. Three different scanning strategies are applied to collect different kind of flow field representation. The total amount of collected scans comprises roughly 142 hours that are sorted in three different atmospheric stability classes. By means of Bayesian inference the model parameters of two of the model components of the DWM model, the deficit in the meandering frame of reference and the wake-added turbulence model are calculated. The methodology allows for a calculation of an uncertainty of the parameters established from a probability distribution. The authors show and discuss the results of the parameter fitting also in comparison to other calibration studies of the DWM model. The final evaluation is conducted by comparing the full DWM implementation with the calibrated submodels to a subset of measurements. The manuscript is in general well written and the different steps of the analysis are well documented and explained. The content is relevant to the scientific discussion of using lidar measurements for model validation/calibration, thus I recommend to publish the manuscript after minor revisions.

My main criticism of the manuscript is the lack of discussion of the generalisation of the results. The authors compare their results of the wake deficit modelling with previous studies and claim e.g. that the model parameters in the IEC standard lead to conservative wake predictions. This might be the case in this specific campaign, but I am missing the evidence that the results are fully transferable from the 190 kW V27 to multi-MW wind turbines. This would imply a fully non-dimensionalized problem. It is not the fault of the study that this can not be proved on a single wind turbine, but the generalisation of the results should be handled with more care. I would have expected that discussion at the end of the manuscript.

We now address this point in a new paragraph in the discussion section, where we point out that further work should evaluate whether the obtained results (for a small-sized turbine) are transferable to modern size rotors. As mentioned by the reviewer, this study cannot answer this point explicitly, however, it provides good insights on the quality (and requirements) of the adopted datasets for ensuring reliable and robust calibration of engineering wake models.

This study shows that the model parameters in the IEC standard lead to conservative wake predictions for turbulence levels above 12%. We now specify this result in the abstract, while it was already mentioned in the conclusions. It also discusses why calibrating wake models using power production data (as done in the IEC model) may lead to inaccurate representations of the wake deficit for varying inflow wind conditions. Further, it shows that the IEC-based wake deficit recovery is basically invariant to ambient turbulence raising from 7% to 16% (see Fig. 9). This is not what we observe from measurements, as shown by the lidar data analysis in this study, and also discussed in the recent work of Reinwardt et al. (2020) [1].

Further, the DWM model is an engineering wake model based on simplistic flow modeling

assumptions. One of these assumptions is that the calibration parameters (e.g., $k_1$ and $k_2$) are universal and independent of the turbine size and inflow wind conditions [2]. However, this assumption might not be valid, as the reviewer mentioned.

Furthermore the final validation of the results compares the calibrated model with the measurements that are used for calibration (even if these are technically not the same measurements as measured in a different period). Thus, it represents rather a verification of the calibration than a model validation.

We called "validation" as we compare model predictions directly with field observations. Also, the validation/comparison of the DWM model-predicted two-dimensional wind speed and turbulence profiles in the FFoR against field observations was not carried out in previous studies.

We rephrased the heading of Sect. 6 from "Validation of the DWM model" to "Validation of the DWM model in the FFoR", in order to specify that we are validating the DWM model-based wake flow field predictions in the FFoR against the lidar data.

Further Comments:

1. Title - As part 2 of this study is only mentioned in the very last sentence of the manuscript and as it includes further measurements, my suggestion is to remove Part I from the title

Part I is now removed as suggested.

2. L10 - We demonstrate that Please edit according to the previous comments on the generalisation of the study

We rephrased the sentence as: "We show that the current DWM-model parameters in the IEC standard lead to conservative wake deficit predictions for ambient turbulence intensities above 12% at the SWiFT site."

3. L42 - load responses mostly Please rephrase.

We rephrased it as: "These three components are presumed to affect wind turbine loading conditions".

4. L46 - still under judgement - still to be assessed.

This has been rephrased as suggested.

5. L71 - and at the DTU ...

We added "the" as suggested.

6. 3.2.1 Atmospheric stability - Can the authors please explain the benefit of sorting the measurements into stability classes for the analysis. as the model has no dependency to atmospheric stability, but just to TI. Why not just sort by TI?

The DWM model's wake deficit formulation in Eq. 1 does not depend on the atmospheric stability but on TI only. However, as the SWiFT campaign provides a comprehensive dataset that allows for characterizing atmospheric stability, we opted for a more detailed analysis of the ambient wind conditions. Such analysis might also be useful for future studies on wake modeling that account for stability effects using the SWiFT dataset.

Further, the DWM model formulation of the wake meandering depends on the spectral properties of the ambient turbulence. Indeed, both TI and the turbulence length scale influence the intensity of the wake meandering. Thus, classifying inflow conditions according to atmospheric stability is necessary when studying meandering dynamics [3].

Finally, atmospheric stability and ambient turbulence are highly correlated, as shown in Table 1, and the adopted classification does not influence the resulting calibration parameters.

7. L249 - T is not a variable in equation (5), virtual potential temperature needs to be introduced

The reviewer refers to the following equation:

$$L = -\frac{u_*^3 T}{kg\overline{w'\Theta_v'}}, \tag{1}$$

where $T$ is the mean surface-layer temperature, and $\Theta_v'$ is the virtual potential temperature. We added "virtual" in the text that was previously missing. In the manuscript, we use the definition of the Obukhov length of Peña et al., (2010) [4] - Eq.(3).

8. 262 - this is sufficient ... Why is this sufficient?

We deleted "sufficient" and rephrased the sentence as: "The dataset collected during *Strategy II* is reported in Table 2 and is used to characterize wake turbulence and meandering under different stability conditions."

9. 442 - These deviations are mainly due ... Can you please elaborate what you mean here?

We rephrased the sentence as: "The largest deviations between predicted and measured deficits are found at shorter distances (2–3D) and are mainly due to the model inadequacy to simultaneously fit all the experimental measurements and experimental uncertainties."

10. Figure 8 - As far as I understood all 4 distances are used for the parameter calibration. In the near wake there is not only a quantitative but also a qualitative disagreement between modeled and measured profiles as the double-Gaussian structure is not measured by the lidar. Have the authors considered to discard the near-wake measurements so they do not disturb the parameter fitting? Also, the uncertainty of the model predictions lie in, I would say most of the plots, outside the measurements. How is that possible? Might the uncertainty be underestimated by the uncertainty quantification approach?

We discard the lidar data at 2D in the fitting process; this is done to improve the quality of the fitting in the far-wake region. As this was not specified in the manuscript, we now add a sentence that describes the adopted procedure (ln. 411).

The uncertainty of the model predictions was not correctly propagated in Fig. 8. The error

originated from an erroneous definition of the cross-correlations matrix of the calibration parameters when propagating uncertainties for plotting purposes. We now re-plot Figs. 8 and 9. We also updated the statistics of the parameter $\sigma_{\epsilon_{def}}$ in Table 4.

11. Figure 9 - This is an interesting figure, but it should be pointed out that the parameters from the other studies were derived from completely different data sets. The different results might also be due to a lack of transferability to larger dimension turbines.

We added a paragraph in the text in Sect. 5.2.1 to better discuss the results.

12. Figure 11 - I am confused by the Figure 11 and the corresponding description in the text. From the text I would expect two versions of the wake-added turbulence in the DWM model, one based on the parameter-fitting of eq. 3 and one based on the parameter fitting of eq. 17.

We now add both wake-added turbulence models from Eqs. 3 and 17.

13. 6.1 Correction for rotor induction effect - It's unfortunate that the measurements had to be conducted in the induction zone of the turbine. The uncertainty introduced by the model correction has to be at least mentioned and preferable quantified.

We mentioned in the text (Ln. 562) that the wind speed is reduced by up to 12% in the induction zone. We now plot the SpinnerLidar-measured statistics of the rotor-effective wind speed ($U_{eff}$) without accounting for induction effects in Fig. 14 for completeness.

14. Figure 14 - Please also show and discuss the results of DWM**, because the DWM* is as far as I understood not the DWM model as it would be applied based on wind measurements at or upstream of the turbine.

We added the predictions from DWM**.

15. L655 - Our result indicate ... Please also here consider the limits to generalise from these results.

We now address the validity of our results in the discussion session.

**References**

[1] Inga Reinwardt, Levin Schilling, Peter Dalhoff, Dirk Steudel, and Michael Breuer. Dynamic wake meandering model calibration using nacelle-mounted lidar systems. *Wind Energy Science*, 5(2):775–792, 2020.

[2] Rolf-Erik Keck, Dick Veldkamp, Helge Aagaard Madsen, and Gunner Chr. Larsen. Implementation of a mixing length turbulence formulation into the dynamic wake meandering model. *Journal of Solar Energy Engineering*, 134(2):021012, 2012.

[3] Ewan Machefaux, Gunner C. Larsen, Tilman Koblitz, Niels Troldborg, Mark C. Kelly, Abhijit Chougule, Kurt Schaldemose Hansen, and Javier Sanz Rodrigo. An experimental and numerical study of the atmospheric stability impact on wind turbine wakes. *Wind Energy*, 19(10):1785–1805, 2016.

[4] Alfredo Peña, Sven-Erik Gryning, and Jakob Mann. On the length-scale of the wind profile. *Quarterly Journal of the Royal Meteorological Society*, 136(653):2119–2131, 2010.

---

## Author Comment (AC2)

**Author response to reviewer 2**

The authors response is shown in red

We thank the reviewer for the valuable comments and suggestions, which we consider very important and help us to sharpen and improve the manuscript. Here our response to each comment.

Overall Impression:

This is a very interesting manuscript that describes the steps to calibrate the DWM and how it compares to the measurement data. The manuscript is very well written and can be easily followed. The analysis is explained in a structured way and the reader can follow the steps carried out by the author. The content is very useful for the scientific discussion to discuss methods to calibrate the DWM, I thus recommend publishing the manuscript after minor revisions.

My main criticisms are the following:

- The manuscript can be shortened as some information's are written multiple times.

- Some steps/ assumptions require additional description to clarify the used approach

- The validation can be more comprehensive by using data used in other studies to show that the newly proposed model is applicable for a wide range of conditions (turbine type, site conditions etc.)

  We answered to this point in a comment below and added a paragraph in the discussion section to better address this aspect.

Comments

1. L145: instantaneous wake radius – 'instantaneous' would indicate that the wake radius is dependent on time. However, to my understanding the averaged N-S is used here. I would rephrase it.

   We deleted 'instantaneous' from the sentence.

2. L160: Is it assumed that the parameter $k_1$ and $k_2$ are independent of wind turbine design and ambient conditions? To my understanding, these parameters are dependent on the assumption that the pressure region ends at $\sim 2D$. Wouldn't the pressure region change at different turbine designs and inflow conditions, such as the atmospheric conditions, shear, or veer, and require a new calibration of $k_1$ and $k_2$? Please add a reference where it states you can assume that these parameters can be considered constant

   The DWM model is an engineering wake model based on simplistic assumptions. In accordance to the DWM model formulation, $k_1$ and $k_2$ parameters are assumed universal, i.e., they do not change with ambient condition or turbine model (see Keck et al. 2012 [1]). Although the pressure region might change at different turbine design and inflow conditions, these effects are not captured in the current DWM model formulation.

We added the reference in the manuscript.

3. L165: In the entire manuscript the focus is on the wake-added turbulence. The first sentence and the equation of the wake turbulence can be removed to shorten the paragraph.

The spatial distribution of the wake turbulence shown in Fig. 13 is derived by linearly summing different turbulence sources as described in Eq. 2. Similarly, the results shown in Fig. 14-b are obtained using the same equation. We also refer to Eq. 2 in Sect. 5.3.1, when isolating the wake-added turbulence term from the total wake turbulence. For these reasons, we keep Eq. 2 and the first sentences describing how wake turbulence is defined in the DWM model.

4. L205: Change the reference Herges et al. 2018, as it refers to Berg et al. 2014 regarding the power output

This has been corrected.

5. L205: Please also add the cut-in and rated wind speed of the turbine.

We added both the cut in and rated wind speed in the sentence.

6. L215: I would rephrase the sentence "The SpinnerLidar has been mounted either in the spinner or on top of the nacelle of a wind turbine." In this study, only results have been used when the SpinnerLidar was installed on the nacelle

We rephrased the sentence as: "In this study, the SpinnerLidar was installed on the nacelle of the *WTGa1* and scanned the rotor wake at a high temporal and spatial resolution so that wake features could be derived."

7. L235: How was the yaw-offset defined? Is it defined as the difference between the nacelle orientation and the wind direction or is it determined using a wind vane on the nacelle?

The yaw-offset is defined as the difference between the nacelle orientation and the wind direction measured at the mast. We added a sentence in the text.

8. L250: Please indicate how the friction velocity is determined

We added: "where $u_* = \sqrt{-\overline{u'w'}}$ is the friction velocity, $\overline{u'w'}$ is the local kinematic momentum flux, ... "

9. L250: According to Pena et al. 2019 the distribution of stability changes with height. Why is the 18m – Sonic used in this study, while the 10m – Sonic is available?

We added a reference to the conference paper (Conti et al., 2020 [2]), where we firstly analyzed the atmospheric stability conditions at the SWiFT site, and explain the reasoning for using the sonic data at 18 m. In that study, we showed that the sonic measurements at 18 m provided the best fit to the polynomial form of Högström [3], which describes the relation between the dimensionless wind shear $\phi_m$ and the dimensionless stability parameter $z/L$ in the surface layer (see Fig. 3-middle in Conti et al., 2020 [2]). This is why we use the sonic measurements at 18 m for characterizing

stability at the site.

10. L260: Please rephrase "sufficient".

We deleted sufficient and rephrased the sentence as: "The dataset collected during *Strategy II* is reported in Table 2 and is used to characterize wake turbulence and meandering under different stability conditions."

11. L270: Is it valid to assume that w=0 at 1D or 2D? At these distances, you are measuring in the near wake region, which is a very complex flow.

We made this assumption for estimating the velocities from the SpinnerLidar measurements. Note that due to the scanning pattern and relatively low opening angles, the lidar will measure just a few percent of w, which is already low even under wake conditions. So, this assumption is made to reduce errors in the u estimates.

Further, the error introduced by the assumption w=0 was quantified in Debnath et al. 2019 [4]. We rephrased the text as: "Considering the small elevation angles and the typical low values of $w$, we assume $w = 0$ [5, 6]. This assumption may introduce an error up to 3% on the reconstructed horizontal wind speed at short distances (1–2D) [4]."

12. L280: How is the longitudinal difference of the points within each scan accounted for, while interpolating onto a 2m-grid, as the SpinnerLidar measures over a sphere?

We projected each point of the rosette pattern within a 2D plane, thus disregarding the 3D property of the scanning pattern.

13. L290: A bivariate Gaussian shape is fitted to obtain the wake center, where the wake center location is within 10% of the lateral bounds of the scanning area. Would this mean that a part of the wake is outside the scanning area, while the fit is being performed? If this is the case is there a reference that shows that you can still accurately estimate the wake center position?

We refer to the work of Doubrawa et al., 2020 [6], which analyzed a subset of the SWiFT dataset and addressed the issue of misfitting a bivariate Gaussian shape to lidar measurements by filtering out scans in which the wake center location is within ∼10% of the lateral bounds of the scanning area.
In the work of Conti et al., 2020 [2], the authors discussed that the wake center locations reached ∼10% of the lateral bounds of the scanning area only at short distances (up to 2.5D) and under unstable conditions for which high-intensity wake meandering were observed.
Both references [2, 6] are provided in the current version of the manuscript.

14. L305: "... under varying stability regimes during strategy I"

This is added now.

15. L315: Is the assumption of no velocity gradient along the beam valid when the measurement is conducted at the edge of the wake?

We added a sentence mentioning that velocity gradients appear when measuring at wake edges and provided a reference that studied this aspect. We added: "Nevertheless, velocity gradients along the lidar beam may appear when measuring at the wake edges, which can introduce errors in the estimated turbulence [7]."

16. L315: L330: Please rephrase "low turbulence (indicate a range)

We added the value of the ambient turbulence.

17. L350: Is strategy II used here to analyse the wake turbulence? (according to the description on p.8) It is unclear to me which data points are used here. According to the description on p.8 strategy II is used. However, Table 2 indicates that there are no measurements in neutral conditions at 7m/s or only one case in unstable conditions at 8m/s

We clarified that in the caption of Fig. 6 as. "Two-dimensional spatial distribution of the horizontal wind velocity variance ($\sigma_U^2$) derived in the MFoR at 2.5 D in the wake, normalized with the $u$-velocity variance of the ambient wind field ($\sigma_{u,amb}^2$) for three 10-min periods characterized by: **(a)** stable, **(b)** near-neutral, **(c)** unstable conditions. Approximately 298 scans of the wake are processed for each 10-min period. The relative ambient wind speed ranges between 6.5 and 8.5 m/s."

According to Table 2, there are 12 (10-min) periods for stable, 5 for near-neutral, and 9 for unstable conditions, given an inflow wind speed ranging between 6.5 and 8.5 m/s.

18. L365: Was it not possible to determine the thrust coefficient in the free field during the measurement period? If it was possible, why was the thrust coefficient of the aeroelastic model used instead of the free-field data?

We had access to a calibrated and validated aeroelastic model of the V27 at the SWiFT campaign and derived the thrust coefficient directly from the model. We also discuss along the manuscript that the major factor influencing the recovery of the wake in the SWiFT dataset is the ambient turbulence, as the recorded inflow wind speeds are below rated and the thrust coefficient is nearly constant. Further, when performing design load calculations the thrust coefficient is typically obtained from the aeroelastic model of the specific turbine [8].

19. L440: The assumption of the near-wake region also introduces uncertainty on the deficit predictions at short distances

We rephrased the sentence as: "The largest deviations between predicted and measured deficits are found at shorter distances (2–3D) and are mainly due to the model inadequacy to simultaneously fit all the experimental measurements and experimental uncertainties. The assumptions adopted to describe the near-wake region also introduce uncertainty on the deficit predictions at short distances [9, 10]."

20. L455: I suggest shifting the comparison of the calibrated DWM and the accompanying figure to section 6 (Validation). Furthermore, I would also show how the calibrated DWM compares to the data set used in the other studies. As the measurement data in this study are used to calibrate the DWM it is no 'surprise' that the DWM has a better performance. The data sets of the other studies could be used as validation cases to show that the newly proposed model is also

applicable for a wide range of conditions.

We opted to separate the calibration and validation of the wake deficit and added turbulence in the MFoR in Sect. 5, and the validation of the fully-resolved wind/turbulence profile in the FFoR in Sect. 6. Note that previous studies have mostly calibrated and validated the prediction in the MFoR. To better differentiate the two sections, we now denote Sect. 5 as "Calibration of the DWM model in the MFoR", and Sect. 6 as "Validation of the DWM model in the FFoR".

One of the main contributions of our study to the DWM model's literature is the high quality dataset utilized to carry out the calibration and validation analyses, which rely on two-dimensional high spatial and temporal resolution measurements of the wake field by a SpinnerLidar. Previous work such as Larsen et al., 2013 [8] and IEC [11] calibrated the DWM model using power production data from an offshore wind farm, so there is no data available describing the spatial distribution of wind/turbulence in the wake. The work of Madsen et al., 20210 [12] and Keck et al., 2015 [9] are based on limited CFD simulations, which do not cover the full range of inflow wind conditions observed at the SWiFT experiment. Reinwardt et al., 2020 [13] calibrated the velocity deficit by using the maximum lidar-derived deficit value. Note that the nacelle lidar used in [13] could only scan the wake horizontally, so any vertical movement of the wake was not captured, which may lead to inaccurate quantification of the velocity deficit, as also discussed by the authors. These are the main reasons why we do not compare our calibrated model with other datasets, but compare previous calibrations with the SWiFT dataset, which is the most comprehensive dataset in the literature. Also, we now added a paragraph in Sect. 5.2.1 to discuss that previous calibrations were conducted on different datasets and turbine sizes in order to explain why such deviations between different calibrations are found in Fig. 9.

Further, this study shows that most of the calibrations (Madsen$_{2010}$, Larsen$_{2013}$, IEC$_{2019}$, and the SpinnerLidar) perform nearly equally well at low turbulence ($< 7\%$) (see Fig. 9). The differences are seen at higher turbulence. One can clearly see that Larsen$_{2013}$ and IEC$_{2019}$ models do not predict any recovery of the wake when the ambient turbulence raises from 7% to 16% (see Fig. 9), which is unexpected. Similar results are also reported in Reinwardt et al., 2020 [13]. As described in the text in Sect. 5.2.1, the erroneous predictions of the Larsen$_{2013}$ and IEC$_{2019}$ models may follow from the utilized calibration procedure that relied on power production data only without any actual evidence of the wake field.

21. L550: "The wind speed is mainly lower than 9 m/s, thus WTGa1 operates below rated power." This sentence can be removed

This has been removed.

22. L555: It is not clear to me which sector of the measurements was corrected for the induction as the wake would not be affected by the induction zone with south-westerly winds (according to Figure 1). Were all the measurements obtained while the wind was coming from the south? Furthermore, I suggest showing the uncertainty of applying this correction and how it compares with measurement data not influenced by the induction zone.

The dataset collected during *strategy III* includes both south and south-westerly winds. However, independently of the incoming wind direction, only the lidar measurements taken across the rotor area of *WTGa2* are scaled by the induction factor of Eq. 18. Thus, for example, there are

only a few 10-min periods for which the wake scanned by the SpinnerLidar does not hit the *WTGa2* due to either south-westerly winds or because *WTGa1* operates purposely under large yaw offsets (see also Herges et al., 2018 [14]). In either case, the lidar data are not corrected by induction. We now specify this in the text.

Also, we now show the resulting rotor-effective wind speed statistics without induction correction in Fig. 14.

23. Either use Figs or Figure to reference a figure.

We use the full-word "Figure" when starting a new sentence, otherwise we use the shorter version "Fig." when in the middle of a sentence. Likewise, we use Figures and Figs.

24. Figure 1: Is the position of WTGa2 correct? In the description, it states that in strategy III a Lidar scan is performed at 5D behind the rotor. This would mean that the Lidar scan is performed at the same position where WTGa2 is located

We re-plotted Fig. 2 (in the manuscript) showing all the 7 scanned distances by the SpinnerLidar from a top view (see Fig. 1 below). Assuming that WTGa1 and WTGa2 are perfectly aligned during operation, then the central points of the rosette pattern are nearly at the same position of WTGa2. However, as the lidar scans within a sphere and the two turbines are rarely aligned, the scan is performed a few meters in front of WTGa2. Nevertheless, as described in Ln. 326 and also in Herges et al., 2019 [15], we discard invalid measurements that occur due to the return from the rotating rotor of WTGa2, among others.

[Figure]

Figure 1: A schematic view of the SpinnerLidar's scanning pattern: **(a)** a front-view at 2.5D in the wake, **(b)** a top-view including all scanned distances behind the *WTGa1*, which is depicted in solid blue lines. The *WTGa2* is also shown.

25. Figure 4: Please indicate the number of lidar scans used to average the velocity deficit

This has been added now.

26. Figure 6: Please indicate the number of lidar scans used to average the variance

This has been added now.

27. Figure 10: No reference to this figure is found.

We refer to this in figure in the text at page 22 Ln: 485.

28. Figure 11: Please also include the profile of the wake-added turbulence with the 'original' analytical formulation.

This is now done.

29. Table 1: Please indicate the wind speed range e.g. [3 - 4] or [2.5 − 3.5].

This is now done.

**References**

[1] Rolf-Erik Keck, Dick Veldkamp, Helge Aagaard Madsen, and Gunner Chr. Larsen. Implementation of a mixing length turbulence formulation into the dynamic wake meandering model. *Journal of Solar Energy Engineering*, 134(2):021012, 2012.

[2] Davide Conti, Nikolay Dimitrov, Alfredo Peña, and Thomas Herges. Wind turbine wake characterization using the SpinnerLidar measurements. *Journal of Physics: Conference Series*, 1618:062040, sep 2020.

[3] U HOGSTROM. Non-dimensional wind and temperature profiles in the atmospheric surface-layer - a re-evaluation. *Boundary-layer Meteorology*, 42(1-2):55–78, 1988.

[4] M. Debnath, P. Doubrawa, T. Herges, L. A. Martínez-Tossas, D. C. Maniaci, and P. Moriarty. Evaluation of wind speed retrieval from continuous-wave lidar measurements of a wind turbine wake using virtual lidar techniques. *Journal of Physics: Conference Series*, 1256(1):012008, 2019.

[5] P. Doubrawa, M. Debnath, P. J. Moriarty, E. Branlard, T. G. Herges, D. C. Maniaci, and B. Naughton. Benchmarks for model validation based on lidar wake measurements. *Journal of Physics: Conference Series*, 1256(1):012024, 2019.

[6] Paula Doubrawa, Eliot W. Quon, Luis A. Martinez-Tossas, Kelsey Shaler, Mithu Debnath, Nicholas Hamilton, Thomas G. Herges, Dave Maniaci, Christopher L. Kelley, Alan S. Hsieh, Myra L. Blaylock, Paul van der Laan, Søren Juhl Andersen, Sonja Krueger, Marie Cathelain, Wolfgang Schlez, Jason Jonkman, Emmanuel Branlard, Gerald Steinfeld, Sascha Schmidt, Frédéric Blondel, Laura J. Lukassen, and Patrick Moriarty. Multimodel validation of single wakes in neutral and stratified atmospheric conditions. *Wind Energy*, 23(11):2027–2055, 2020.

[7] Alexander Raul Meyer Forsting, Niels Troldborg, and Antoine Borraccino. Modelling lidar volume-averaging and its significance to wind turbine wake measurements: Paper. *Journal of Physics. Conference Series*, 854(1):012014, 2017.

[8] Torben J. Larsen, Helge Aagaard Madsen, Gunner Chr. Larsen, and Kurt Schaldemose Hansen. Validation of the dynamic wake meander model for loads and power production in the egmond aan zee wind farm. *Wind Energy*, 16(4):605–624, 2013.

[9] Rolf Erik Keck, Martin De Maré, Matthew J. Churchfield, Sang Lee, Gunner Larsen, and Helge Aagaard Madsen. Two improvements to the dynamic wake meandering model: Including the effects of atmospheric shear on wake turbulence and incorporating turbulence build-up in a row of wind turbines. *Wind Energy*, 18(1):111–132, 2015.

[10] Ewan Machefaux, Gunner C. Larsen, Tilman Koblitz, Niels Troldborg, Mark C. Kelly, Abhijit Chougule, Kurt Schaldemose Hansen, and Javier Sanz Rodrigo. An experimental and numerical study of the atmospheric stability impact on wind turbine wakes. *Wind Energy*, 19(10):1785–1805, 2016.

[11] International Standard IEC61400-1: wind turbines—part 1: design guidelines, Fourth; 2019. Standard, IEC, 2019.

[12] Helge Aagaard Madsen, Gunner Chr. Larsen, Torben J. Larsen, Niels Troldborg, and Robert Flemming Mikkelsen. Calibration and validation of the dynamic wake meandering model for implementation in an aeroelastic code. *Journal of Solar Energy Engineering*, 132(4):041014, 2010.

[13] Inga Reinwardt, Levin Schilling, Peter Dalhoff, Dirk Steudel, and Michael Breuer. Dynamic wake meandering model calibration using nacelle-mounted lidar systems. *Wind Energy Science*, 5(2):775–792, 2020.

[14] T. G. Herges, J. C. Berg, J. T. Bryant, J. R. White, J. A. Paquette, and B. T. Naughton. Detailed analysis of a waked turbine using a high-resolution scanning lidar. *Journal of Physics: Conference Series*, 1037(7):072009, 2018.

[15] T. G. Herges and P. Keyantuo. Robust lidar data processing and quality control methods developed for the swift wake steering experiment. *Journal of Physics: Conference Series*, 1256(1):012005, 2019.